# Intelligent image-based in situ single-cell isolation

Csilla Brasko[1], Kevin Smith[2,3], Csaba Molnar[4], Nora Farago[1,4,5], Lili Hegedus[4], Arpad Balind[4], Tamas Balassa[4], Abel Szkalisity[4], Farkas Sukosd[1], Katalin Kocsis[1], Balazs Balint [6], Lassi Paavolainen[7], Marton Z. Enyedi[4], Istvan Nagy [4,6], Laszlo G. Puskas[4,5], Lajos Haracska[4], Gabor Tamas[1] & Peter Horvath[4,7]

Quantifying heterogeneities within cell populations is important for many fields including cancer research and neurobiology; however, techniques to isolate individual cells are limited. Here, we describe a high-throughput, non-disruptive, and cost-effective isolation method that is capable of capturing individually targeted cells using widely available techniques. Using high-resolution microscopy, laser microcapture microscopy, image analysis, and machine learning, our technology enables scalable molecular genetic analysis of single cells, targetable by morphology or location within the sample.

---

[1] University of Szeged, Szeged, Hungary Közép fasor 52, 6726 Szeged Hungary. [2] School of Computer Science and Communication, KTH Royal Institute of Technology, Lindstedtsvägen 3-5, 10044 Stockholm Sweden. [3] Science for Life Laboratory, Tomtebodavägen 23A, 17121 Solna Sweden. [4] Biological Research Centre of the Hungarian Academy of Sciences, Temesvári krt. 62., 6726 Szeged Hungary. [5] Avidin Biotechnology Ltd, Alsó Kikötő sor 11, 6726 Szeged Hungary. [6] SeqOmics Biotechnology Ltd, Vállalkozók útja 7, 6782 Mórahalom Hungary. [7] Institute for Molecular Medicine Finland (FIMM), University of Helsinki, Tukholmankatu 8, Helsinki 00014, Finland. Csilla Brasko, Kevin Smith, Csaba Molnar, Gabor Tamas and Peter Horvath contributed equally to this work. Correspondence and requests for materials should be addressed to P.H. (email: horvath.peter@brc.mta.hu)

Much of our current understanding of biology is built upon population-averaged measurements, including many models for cellular networks and signaling[1]. However, measurements averaging the behavior of large populations of cells can lead to false conclusions if they mask the presence of rare but critical subpopulations[2]. It is now well recognized that heterogeneities within a small subpopulation can carry important consequences for the entire population. For example, genetic heterogeneity plays a crucial role in drug resistance and the survival of tumors[3]. Even genetically homogeneous cell populations possess large degrees of phenotypic cell-to-cell variability due to individual gene expression patterns[4]. To better understand biological systems with cellular heterogeneity, we increasingly rely on single-cell molecular analysis methods[5]. However, single-cell isolation, the process by which we target and collect individual cells for further study, is still technically challenging and lacks a perfect solution.

A number of isolation methods are capable of collecting cells based on certain single-cell properties in a high-throughput manner, including fluorescence-activated cell sorting (FACS), immunomagnetic cell sorting, microfluidics, and limiting dilution[6,7]. However, these harvesting techniques disrupt and dissociate the cells from the microenvironment, and they are incapable of targeting the cell based on location within the sample or by phenotypic profile. In contrast, micromanipulation and laser capture microdissection[8] (LCM) are microscopy-based alternatives that directly capture single cells from suspensions or solid tissue samples. They can target cells by location or phenotype, and this contextual information can provide important insights when interpreting data from genetic analysis. LCM and micromanipulation methods can isolate specific subpopulations without substantial disruption of the tissue while limiting contamination (e.g., from chemical treatments needed for FACS). This is an important advantage for assaying single-cell gene expression and molecular processes. Recently, other single-cell isolation techniques have been introduced to perform mass spectrometry on single cells[9]. However, all these methods have a crucial limitation—they require manual operation to choose cells for isolation and to precisely target and extract them. These human-operated steps are error-prone and laborious, which greatly limits capacity.

We developed a technique to increase the accuracy and throughput of microscopy-based single-cell isolation by automating the target selection and isolation process. Computer-assisted microscopy isolation (CAMI) combines image analysis algorithms, machine-learning, and high-throughput microscopy to recognize individual cells in suspensions or tissue and automatically guide extraction through LCM or micromanipulation. To demonstrate the capabilities of our approach, we conducted three sets of experiments that require targeted single-cell isolation to collect individual cells without disturbing their microenvironment. We show that CAMI-selected cells can be successfully used for digital PCR (dPCR) and next-generation sequencing through these experiments.

## Results

**The CAMI system.** A diagram summarizing CAMI technology is provided in Fig. 1. During preparation, samples are collected in variable formats etched with registration landmarks (Supplementary Note 1), and potentially treated with compounds according to the assay (Fig. 1a). Samples may come from tissue or cell cultures, and they are imaged with an automated high-throughput microscope (Fig. 1b). Images from the microscope are sent to our image analysis software that uses state-of-the-art algorithms to correct illumination, identify and segment cells (even in cases of overlap, Supplementary Note 2)[10], and extract multiparametric cellular measurements[11] (Fig. 1c). Advanced Cell Classifier software[12] trains machine-learning algorithms to automatically recognize the cellular phenotype of every cell in the sample based on their extracted properties (Fig. 1d), and these data along with the location and contour of each cell are sent to our interactive online database computer-aided microscopic isolation online (CAMIO; Fig. 1e). CAMIO provides an interface to approve the cells chosen to be extracted. If the user wishes, he/she may add or remove cells, or correct mistakes in the contour and classified phenotype. Selected cells are then extracted by micromanipulation or laser microdissection combined with a catapulting system (Fig. 1f) and collected in a microtube or high-throughput format for molecular characterization such as sequencing or dPCR (Fig. 1g). The software components we developed to support this technology are freely available (Supplementary Software).

As a proof of principle, we conducted three sets of experiments to demonstrate the capabilities of the technology to target, isolate, and analyze individual cells without disturbing their microenvironment (Fig. 2). These experiments were chosen because they could not have been analyzed using conventional automated isolation techniques (e.g., FACS), and alternative solutions would have required laborious manual operation.

**Cell selection by phenotype validated by dPCR.** First, we check whether immunofluorescent-labeled cells selected using machine-learning in CAMI corresponded to mRNA quantification in individual neurons extracted from 10 µm thick sections of the rat cerebral cortex. To accomplish this, we applied fluorescent labels to tissue fixed in 4% paraformaldehyde using immunohistochemistry with nNOS antibodies (Fig. 2a). Then, we automatically targeted and extracted individual cells that were predicted to belong to two phenotypic categories using CAMI technology (Fig. 2b). Cells that were most confidently predicted to be nNOS-expressing interneurons and non-labeled pyramidal cells were selected and isolated using laser microdissection (Fig. 2c). These individual cells were then catapulted and collected in PCR tubes containing SingleCellProtect stabilization and lysis buffer and directly used for cDNA conversion (Fig. 2d). The cDNA mixture was divided into two parts and used for single-cell dPCR to measure nNOS and RS18 gene expression for each extracted cell[13]. The dPCR results confirm that single nNOS-expressing neurons were reliably separated from nearby cells of other types within the same tissue (Fig. 2e). We also checked that the RNA did not significantly degrade by confirming that the number of transcripts of RS18, a housekeeping gene used as a control and for normalization between cells, matched values previously observed using live-cell aspirates with good fidelity[13,14] (Supplementary Data 1).

**Whole-transcriptome sequencing of pyramidal cells.** Next, we applied CAMI to target pyramidal cells for isolation from the same cortex sections. Using an online cell isolation tool we developed CAMIO, we automatically identified pyramidal cells based on morphological image features and selected cells specifically from layers L2 and L3 of the somatosensory cortex. The cells were extracted using LCM, pooled in SingleCellProtect buffer, and amplified using a REPLI-g WTA Single Cell Kit that contains an optimized Phi 29 polymerase and uses multiple displacement amplification technology. After quality control, we prepared sequencing libraries from the purified cDNA, sequenced the fragments on an Ion Torrent PGM, and recorded a list of gene expression. The experiment was repeated for three biological replicates (50, 50, and 300 cells), and the whole-transcriptome

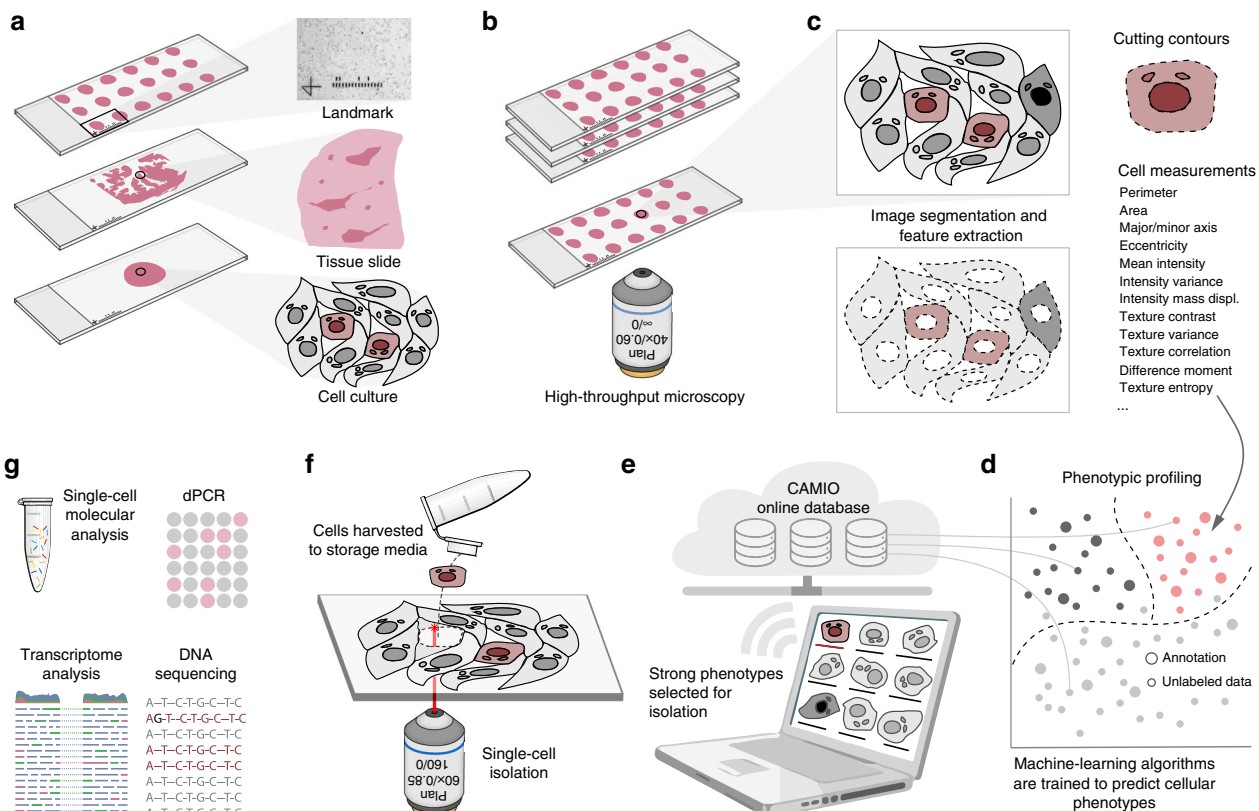

**Fig. 1** Summary of computer-assisted microscopy isolation technology. **a** Tissue or cultured samples are prepared in a variety of formats, etched with registration landmarks, and treated according to the assay. **b** Samples are imaged with an automated high-throughput microscope. **c** Image analysis software applies algorithms to correct illumination, identify and segment cells (even in cases of overlap)[10], and extract multiparametric cellular measurements. Our software automatically defines a cutting contour using these data. **d** Advanced Cell Classifier software trains and optimizes machine-learning algorithms to automatically recognize cellular phenotypes based on extracted properties. **e** The raw images and analysis data are sent to our interactive online database, which provides an interface to review and select imaged cells. Cells exhibiting strong phenotypes are recommended for extraction. The user can add or remove cells or correct mistakes on the contour and classified phenotype prior to extraction. **f** Selected cells are extracted by micromanipulation or laser microdissection combined with a catapulting system and collected in a microtube or high-throughput format. **g** Outside the CAMI workflow, the collected cells can be molecularly characterized (e.g., digital PCR or next-generation sequencing)

profiles were compared (Fig. 2f). A comparison of the profiles revealed high correlations (Pearson's R) and high overlap in the top-100 expressed genes between the replicates (Supplementary Data 2, Supplementary Data 3). In a similar procedure, 50 astrocytes were also collected and sequenced, revealing negligible correlation with the pyramidal cells (Fig. 2f, Supplementary Data 2). This experiment shows that it is possible to automatically collect populations of a distinct type of cell from a specific region of fixed tissue in a high-throughput manner, and to perform reproducible whole-transcriptome sequencing using CAMI extraction.

**Identification of upstream regulators by phenotyping.** Last, we demonstrate that CAMI technology can provide a highly sensitive and cost-effective alternative to RNA interference (RNAi) library screens to uncover novel gene functions. While RNAi knock-downs test one gene at a time—measuring population responses (~20,000 experiments for a genome-wide library)—CAMI technology can be used to select individual cells from a mixture of stably silenced cell lines. Pooled cells exhibiting interesting phenotypes can be collected for further analysis, and the cell's silenced gene can be identified. The DNA of extracted cells is sequenced using universal primers flanking the specific silencing short hairpin RNA (shRNA)-coding region present in each cell of the library. As a proof of concept, we followed this approach to identify both known and novel genes involved in the response to

DNA damage. We prepared a mixture of single shRNA-expressing stable human embryonic kidney cell lines (limited to 10 cell lines in our study). DNA damage was induced in the cells through UV exposure. In normal cells, this results in the recruitment of DNA repair proteins to the damage site and the formation of nuclear foci[15]. A fluorescent marker indicating polymerase η expression allowed us to visualize the formation of foci as spots within the nucleus (Fig. 2g). In the absence of upstream regulators, recruitment of repair proteins to the damage sites is prevented, resulting in a homogeneous expression of polymerase η (Fig. 2h). Using CAMI, we automatically identified 150 foci-forming and 150 homogeneous cells, captured them, and sequenced their shRNA-coding DNA region using next-generation sequencing (NGS). Our results confirm the identification of previously published upstream regulators of polymerase η (SPARTAN, BRCA2, and RAD18)[16–18], and identified RAD52 and FANCA as promising new potential regulators (Fig. 2i).

## Discussion

LCM has been around for nearly 20 years[19], yet only now have the technologies matured sufficiently and computational techniques become sophisticated enough to support targeted automatic, environment-preserving, high-throughput single-cell isolation as we propose. Computer-driven automation increases throughput over manual techniques by orders of magnitude (from several hundred to over a thousand cells per day with CAMI, see

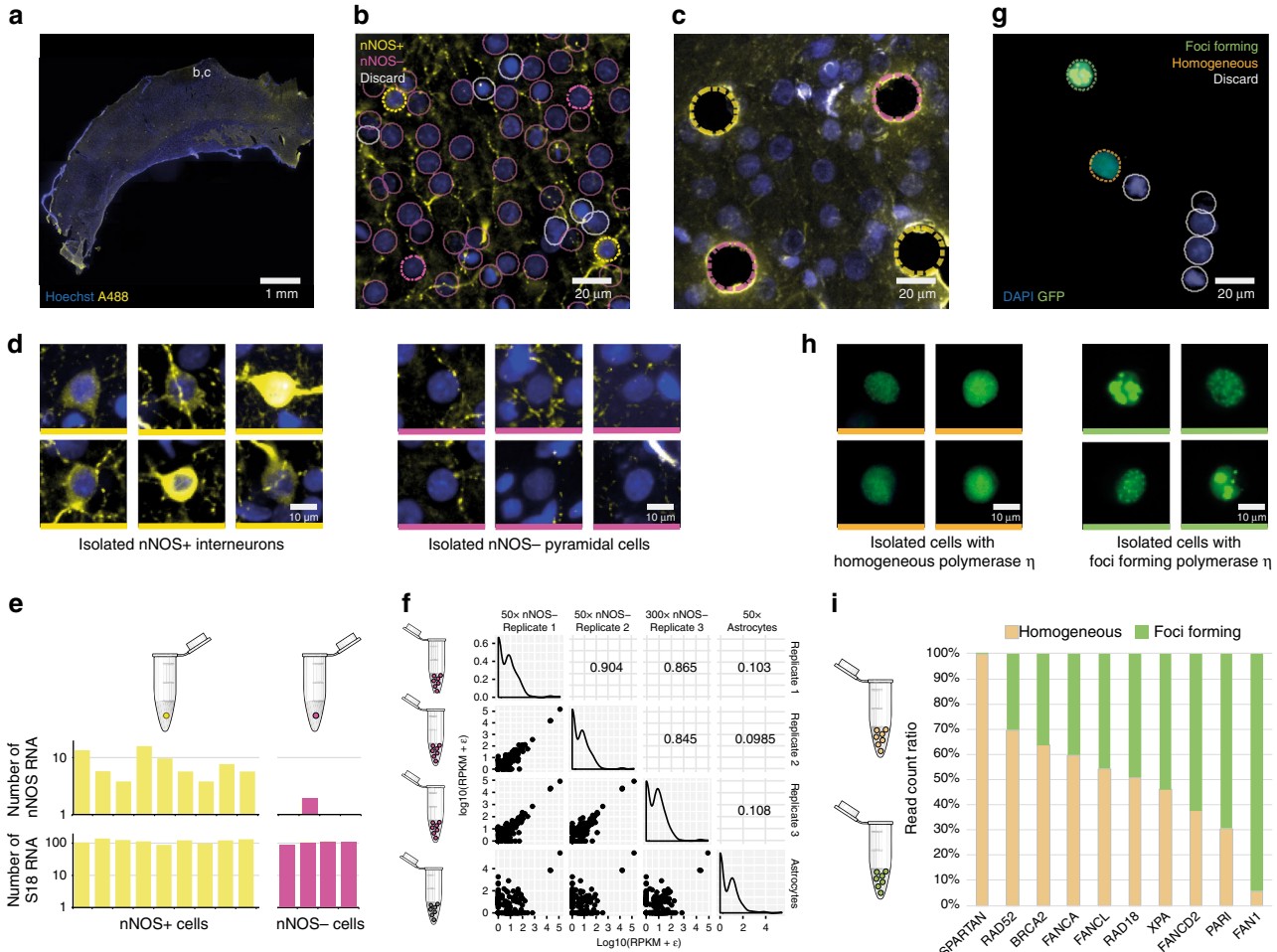

**Fig. 2** Computer-assisted microscopy isolation (CAMI) opens the door to new types of high-throughput single-cell molecular analysis through non-disruptive collection of individual cells from fixed tissue and selection of cells by phenotypic morphology or location. **a** Coronal sections of rat brain labeled with mouse-anti-NeuN antibody (blue) and rabbit anti-nNOS antibody (yellow) were imaged with a high-throughput microscope. **b** High-resolution detail of a region of the somatosensory cortex indicated in **a**. Outlines show nuclear segmentations and phenotype classifications predicted by our software. Cells outlined in yellow are predicted to be nNOS+, cells outlined in magenta are nNOS−, and gray indicates cells that should be discarded (e.g., due to artifacts). Dotted lines indicate cells that were targeted for extraction. **c** The same region after extracting two nNOS+ and two nNOS− cells. **d** Individual cells automatically selected and extracted using CAMI, nNOS-expressing interneurons on the left and nonexpressing cells on the right. **e** Expression levels measured by dPCR show that CAMI reliably separates cells. Cells identified as nNOS+ show significantly higher expression ($7.96 \pm 0.48$) than those identified as nNOS− ($0.48 \pm 0.95$), two-sampled $t$-test $p = 0.0061$. Expression levels of housekeeping gene S18 did not vary significantly between cells identified as nNOS+ ($116.37 \pm 16.54$) and nNOS− ($103.98 \pm 10.29$), two-sampled $t$-test $p = 0.1992$. **f** Whole-transcriptome gene expression profiles of nNOS− cells (two 50-cell replicates and one 300-cell) and astrocytes (50 cells) extracted by CAMI and sequenced by Ion Torrent PGM. Analysis reveals strong correlations (Pearson's R) between the nNOS− replicates, and weak correlations between the astrocytes and nNOS− cells. **g** CAMI also enables a novel, cost-effective alternative to RNAi screening. Cells with interesting phenotypes are identified and extracted from mixed populations of stable shRNA-expressing silenced cell lines. After UV exposure, cells normally recruit polymerase η to repair DNA damage, which is visualized as foci by our green fluorescent marker. Absence of an upstream regulator can disrupt the foci formation and lead to homogeneous polymerase η expression. **h** CAMI identified 150 foci-forming and 150 homogeneous cells and extracted them. **i** Extracted cells were sequenced using next-generation sequencing (NGS). The ratio between the two populations revealed known upstream regulators of polymerase η (BRCA2, RAD18, and SPARTAN) and identified promising new regulators, Rad52 and FANCA

Supplementary Note 3, compared to 10 with patch-clamp harvesting), and microscopy-based isolation boasts several advantages over conventional high-throughput isolation techniques. These include non-disruptive collection of individual cells from fixed tissue or cell culture and selection of cells based on phenotypic morphology or location within the tissue. The throughput, precision, and versatility of CAMI enable new modes of highly reproducible molecular analysis and make it an attractive technique to drive new discoveries, for example, through alternative RNA and CRISPR/Cas9-screening approaches or through clinical applications using fresh or archived tissue samples.

## Methods

**Set-up.** As a first step for every experiment, we etched $50 \times 50\,\mu m$ landmarks into poly-L-lysine-coated slides (one landmark per slide) using a microdissection microscope (Zeiss Axio Observer microscope with PALM MicroBeam manipulator). The landmarks are easily recognized by software and serve as an absolute zero position to register image data between microscopes. The landmarks were designed to indicate the orientation in order to avoid any errors due to rotation of the coordinates (Supplementary Note 1). An image of the landmark is also acquired and stored. Optionally, unique barcodes may also be etched into the slide to identify samples.

**Tissue preparation**. Male Wistar rats (between 200 and 350 g) were anesthetized by inhalation of 2-Bromo-2-chloro-1,1,1-trifluoroethane followed by intraperitoneal administration of 1 ml of 4% chloral hydrate per 100 g of body weight. Animals were then transcardially perfused with ice-cold saline for 2–4 min (10 ml/ 1 min) followed by 4% paraformaldehyde (PFA) made up in 0.1 M phosphate buffer (PB, pH = 7.4) for 10 min. Coronal sections of 10 μm thickness were cut with a Leica vibratome (Leica, VT 1000 S). All procedures were performed with the approval of the University of Szeged and in accordance with the National Institutes of Health Guide for the Care and Use of Laboratory Animals.

**Immunohistochemistry**. Rat coronal sections were washed twice with 0.1 M PB for 10 min, followed by two washes with Tris-buffered saline (TBS, pH = 7.4) for 10 min. Sections were blocked for 2 h in a solution containing 20% normal horse serum and 0.2% Triton-X-100 made up in TBS. The sections were then incubated with primary antibodies to detect markers for neuronal cell populations including a mouse-anti-NeuN antibody (1:2000, MAB377, Chemicon, Temecula, CA) and a rabbit anti-nNOS antibody (1:1200, 160870 Cayman Chemical Company, Ann Arbor, MI). The antibodies were diluted in TBS and incubated for 2 days. After incubation, sections were washed four times with TBS for 10 min, and secondary antibodies including donkey anti-rabbit Alexa Fluor 488 (711-545-152, Jackson ImmunoResearch Laboratories, West Grove, PA), donkey anti-rabbit Cy3 (711-165-152, Jackson ImmunoResearch Laboratories), and donkey anti-mouse Alexa Fluor 568 (A10037, Thermo Fisher) were applied in 1:400 dilution and incubated for 2.5 h at room temperature. During the last 30 min of the incubation, Hoechst blue (Sigma, B2261) was added in 1 μg/ml concentration. Finally, sections were washed 3× with TBS for 10 min and then washed 2× with 0.1 M PB for 5 min before mounting in vectrashield (H-1000 Vector Laboratories, Burlingame, CA).

**Imaging set-up and acquisition**. Prior to extraction, a high-throughput screening campaign was performed using an automated imaging system (Operetta, PerkinElmer, Germany) allowing us to automatically analyze thousands of cells and to select the best examples for isolation. A 20× long working distance objective with 0% overlap was used to collect 1200 images with two fluorescent channels: Hoescht 333 and Alexa 568. The system we propose is compatible with any open format microscope where image position and pixel size can be measured, and has been successfully tested with a confocal slide scanner (Pannoramic Confocal, 3DHistech, Hungary) and a laser-scanning confocal microscope (FV 1000, Olympus, Japan) using 20× water and 40× oil emerging objectives. Alternatively, manual cell selection can be performed directly using the dissection microscope. However, throughput is significantly reduced.

**Image analysis and pattern recognition**. We developed a software pipeline to precisely outline every cell from the screen and to predict its phenotypic class. This software allows us to quickly visualize and select the best cell candidates from relevant subpopulations for isolation. The pipeline is composed of three steps: pre-processing, segmentation and feature extraction, and classification. The pre-processing step corrects artifacts due to uneven illumination present in the images using a quasi-newtonian optimization technique[20]. In the segmentation step, cells were outlined and properties were extracted using CellProfiler software[11] with custom pipelines. If individual cells were well separated, the default nuclei segmentation method was used: a seed-based adaptive Otsu thresholding. For cells in close proximity to one another, this method often fails. To overcome this, we used a two-step approach that first identifies nucleus centers using an *à trous* wavelet transform[21] and then expands the seeds to fit the boundaries using either CellProfiler secondary objects or high-order active contours[10] in the case of overlapping cells (Supplementary Note 2). This method allowed us to reliably identify cells with overlapping nuclei, which are typically discarded from molecular analysis. Custom CellProfiler modules implementing these methods and the pipelines used are provided (Supplementary Software). Nucleus segmentations were used to construct a polygon approximation of a 3 μm around the nucleus. This defines the cutting regions for isolation. It ensures that the laser does not destroy molecular information from the nucleus and also minimizes contamination from extracellular sources. After the segmentation step is complete, 92 single-cell properties describing the intensity, texture, and shape of the nuclei were extracted using CellProfiler and stored in the Advanced Cell Classifier (ACC) format[12].

We used supervised machine-learning algorithms to predict the phenotypic class for every cell in the screens based on the extracted features. Using Advanced Cell Classifier software[11], segmented objects were labeled according to their phenotypes. Using these data as a training set, ACC was used to train several machine-learning models using multiple methods to predict phenotypic class of all cells; 10-fold cross-validation was used to select the best-performing model. A random forest classifier achieved 91% cross-validation accuracy and was trained using every annotation (Supplementary Figure 1). It was then used to predict the phenotypic class for every cell. ACC software with modules to upload single-cell information for selected subpopulations to an online repository is included as Supplementary Software.

**Single-cell online repository and selection tool**. Cell phenotype predictions are ranked by confidence, and the 200 cells with highest confidence for the interneuron

and pyramidal phenotype classes were automatically uploaded to CAMIO, an online single-cell data repository and selection tool we developed (Supplementary Software). The purpose of this tool is to visualize individual cells and facilitate the selection of appropriate candidates for isolation. Individual cells are displayed, organized by experiment and phenotypic class. Cells can be selected for isolation manually or through manual verification. Selected cells are sent instantly to the single-cell isolation device. The CAMIO interface allows the user to verify and correct the proposed cutting regions for each cell. It also records the location of the etched landmark relative to each object. The CAMIO interface is shown in Supplementary Figure 2, and an online read-only version of the system can be tested at https://camio-webapp.herokuapp.com/.

**Image coordinate registration between microscopes**. To register data between microscopes, the landmark etched in each sample slide is automatically detected by our software using two-dimensional cross-correlation. The landmark location is used as the zero position and orientation reference to transform data from one microscope to another. The offset between the orientation landmark and the microscope coordinate system is recorded in the source microscope (high-throughput microscope) and recorded. It is also measured in the target microscope (laser microdissection microscope). With this information, coordinates defining the cutting region for a cell can be transformed from the source image coordinates to the target microscope coordinates using the following relation

$$(x^2, y^2)^T = (y^1, x^1)^T - (y^1_{\text{off}}, x^1_{\text{off}})^T + (x^2_{\text{off}}, y^2_{\text{off}})^T$$

where $x^1$ and $y^1$ are the coordinates in the source microscope, $x^1_{\text{off}}$ and $y^1_{\text{off}}$ are the origin offsets in the source microscope, and $x^2_{\text{off}}$ and $y^2_{\text{off}}$ are the origin offsets in the target microscope. By applying this transform, contours of cells from the high-throughput microscope and CAMIO can be registered in the laser microdissection microscope.

**Single-cell isolation**. To prevent contamination, a custom-designed closed hood was mounted on the isolation microscope and a UV sterilizer was built in (UVR-M Biosan) that was run before every experiment for 30 min. Temperature in the hood and laboratory was 20 °C, and humidity was kept at 50–60% to prevent sample drying.

After cells were selected for isolation using the CAMIO online tool, samples were hydrated with 0.1 M PB. The tissues were initially overhydrated. Immediately prior to cell isolation, the liquid was entirely removed from the surface. This practice allowed a more flexible schedule when cutting. The cutting path for each cell was provided by CAMIO. As a last step before each cell was extracted, we acquired an image of the specimen in situ to document the cell before isolation. This allowed us to perform quality control and refer to the source image when examining results from further analysis. A Zeiss PALM laser microdissection microscope was used for isolation with a 63× LCM-compatible magnification objective (LD Plan-Neofluar, 63×). The cutting was performed using the ultraviolet (337 nm) N2 laser microbeam system of Zeiss PALM, emitting 3 ns pulses. The laser-cutting speed was 1% (~ 4.7 μm/s), and cutting time ranged between 10 and 20 s per cell, depending on the contour of the cell and stage velocity. The cutting energy varied between 36 and 48 μJ depending on the glass thickness. By keeping the laser pulses short and low-power, we promoted a "cold cutting" that is less harmful to the samples.

Isolated cells were pressure-catapulted into PCR tube caps containing 4 μl SingleCellProtect™ (Avidin Ltd., Szeged, Hungary) buffer media facing downward for storage. To avoid dripping and evaporation of the media, microtubes were kept at −20 °C and catapulting was performed when the buffer transitions from frozen to liquid state (between 10 and 20 s after removal from the fridge). After collection, the tubes were closed and immediately stored at −80 °C.

**Single-cell reverse transcription and dPCR of rat cortical neurons**. Reverse transcription of individual microdissected cells was carried out in two steps. The first step was performed for 5 min at 65 °C in a total reaction volume of 7.5 μl containing the cell captured in 4 μl SingleCellProtect™ (Avidin Ltd., Cat.No.: SCP-250), 0.45 μl TaqMan Assays (Thermo Fisher), 0.45 μl 10 mM dNTPs (Thermo Fisher, Cat.No.: 10297018, 1.5 μl 5× first-strand buffer, 0.45 μl 0.1 mol/l DTT, 0.45 μl RNase inhibitor (Thermo Fisher, Cat.No.:N8080119), and 100 U of reverse transcriptase (Superscript III, Thermo Fisher, Cat.No.: 18080055). The second step of the reaction was carried out at 55 °C for 1 h, and then the reaction was stopped by heating at 75 °C for 15 min. The reverse transcription reaction mix was stored at −20 °C until PCR amplification.

For dPCR analysis, the reverse transcription reaction mixture (7.5 μl) was divided into two parts: 6 μl was used for amplification of the gene of interest and 1.5 μl cDNA was used for amplifying the housekeeping gene, RS18. Template cDNA was supplemented with nuclease-free water to a final volume of 8 μl. TaqMan Assays (2 μl; Thermo Fisher), 10 μl OpenArray Digital PCR Master Mix (Thermo Fisher, Cat.No.: 4458095), and nuclease-free water (3 μl) were mixed to obtain a total volume of 20 μl, and the mixture was evenly distributed on four subarrays (256 nanocapillary holes) of an OpenArray plate by using the OpenArray autoloader. Processing of the OpenArray slide, cycling in the OpenArray NT cycler, and data analysis were done as previously described[22]. For our dPCR

protocol amplification, reactions having CT values less than 23 or greater than 33 were considered primer dimers or background signals, respectively, and excluded from the data set.

The following Taqman Assays were used: RS18 (Thermo Fisher, Cat.No.: 4331182, Rn01428913_gH), NOS1 (Thermo Fisher, Cat.No.: 4331182, Rn00583793_m1), NPY (Thermo Fisher, Cat.No.: 4331182, Rn01410145_m1).

**Whole-transcriptome sequencing of rat cortical neurons**. For RNA and subsequent cDNA amplification, REPLI-g WTA Single Cell Kit (Qiagen, Cat.No.: 150063) was used with the Amplification of Total RNA from Single Cells' protocol according to the manufacturer's guidelines with the exception that 3 μl Lysis buffer was added to 8 μl 1× SingleCellProtect solution containing either 50 or 300 collected cells (three replicates were collected—two with 50 cells and one with 300 cells, denoted 50× nNOS—Replicate 1, 50× nNOS—Replicate 2, and 300× nNOS —Replicate 3. In addition, 50 astrocytes were collected, denoted 50× Astrocytes). All subsequent steps were performed as described in the protocol manual. The quality and quantity control of cDNA pools were performed on TapeStation using genomic DNA ScreenTape and Reagents (Agilent Technologies, Cat.No.: 5067-5365 and 5067-5365) and Qubit using dsDNA High-Sense assay (Thermo Fisher, Cat.No.: Q32854), and were purified using Agencourt AMPure XP magnetic beads (Beckman Coulter, Cat.No.: A63881). Fragment libraries were constructed from purified cDNA using NEBNext Fast DNA Fragmentation & Library Prep Set for Ion Torrent (New England Biolabs, Cat.No.: E6285) according to the manufacturer's instructions. Briefly, cDNA was enzymatically digested, and the fragments were end-repaired; the fragmentation time was adjusted to cDNA quality and quantity (generally 5–8 min of fragmentation). Fragmented cDNA pools were purified with Agencourt AMPure XP magnetic beads. Purified fragments were end-repaired, Ion Xpress Barcode Adaptors (Thermo Fisher, Cat.No.: 4474521) were then ligated and the template fragments size-selected using AMPure beads. Adaptor-ligated fragments were PCR-amplified, cleaned-up using AMPure beads, quality-checked on D1000 ScreenTape and Reagents using TapeStation instrument (Agilent Technologies, Cat.No.: 5067-5582 and 5067-5583), and finally quantified using Ion Library TaqMan Quantitation Kit (Thermo Fisher, Cat.No.: 4468802). The library templates were processed for sequencing using the Life Technologies Ion OneTouch protocols and reagents. Library fragments were clonally amplified onto Ion Sphere Particles (ISPs) through emulsion PCR and then enriched for template-positive ISPs. More specifically, Ion PGM emulsion PCR reactions utilized the Ion OneTouch 200 Template Kit (Thermo Fisher, Cat.No.: 4480974), and emulsions and amplification were generated using the Ion OneTouch System (Thermo Fisher). Enrichment was completed by selectively binding the ISPs containing amplified library fragments to streptavidin-coated magnetic beads, removing empty ISPs through washing steps, and denaturing the library strands to allow for collection of the template-positive ISPs. For all reactions, these steps were accomplished using the ES module of the Ion OneTouch System. Template-positive beads were deposited onto the Ion 318 chips (Thermo Fisher, Cat.No.: 4484354); finally, sequencing was performed with the Ion PGM Hi-Q Sequencing Kit (Thermo Fisher, Cat.No.: A25592) on Ion Torrent PGM instrument generating between 2.9 and 5.3 million reads per sample.

**Ion Torrent PGM sequencing data processing and expression analysis**. The PGM sequencing data were processed using Genomics Workbench ver 9.0.1 (CLC Bio). Raw sequencing data were trimmed by removal of low-quality (quality limit: 0.05) and short (length limit: 40 bases) sequences so that only high-quality sequences were used in further analysis. Sequences were mapped on the Rattus norvegicus 6.0 genome (Rnor_6.0) using the CLC RNA-Seq algorithm, allowing mapping to intergenic regions, using default parameters except for the following: minimum alignment length 80%, minimum similarity 80% with the maximum number of hits for a read set to 30. Total read counts were used as a measure of gene expression in all samples.

The level of correlation between the biological replicates was determined by using the Pearson's product–moment correlation coefficient (PCC) which infers the linear relationship between two data sets based on the covariance and SD from the expression values. These values computed between each nNOS− replicate and the astrocytes are provided in Fig. 2f.

**Sample preparation to detect subnuclear foci formation in human cells**. HEK293 cells stably expressing different shRNAs were harvested in Dulbecco's modified Eagle's medium (Sigma, Cat. No. D6429) supplemented with 10% fetal bovine serum (Gibco, Cat. No. 10270) and 300 μg/ml G418 (Biochrom, Cat. No. A291-25) at 37 °C. The cells were transfected with GFP-polymerase η-expressing plasmid using the Lipofectamine 2000 transfection reagent (Invitrogen, Cat. No. 11668). Cells were plated in six-well plates 24 h before transfection. Then, the growth media was removed and changed to 1.5 ml OptiMEM per well. An amount of 3 μg plasmid DNA and 5 μl Lipofectamine 2000 reagent were used for each well. Both the DNA and the transfection reagent were diluted in 250–250 μl OptiMEM, mixed by vortexing, and incubated for 5 min. After mixing the two tubes, the solution was further incubated for 20 min, added to the cells dropwise and incubated for 4 h, and then the transfection media were removed and changed to 3 ml fresh growth media.

After 48 h had elapsed since transfection, cells were exposed to 20 J/m² UVC light to induce DNA damage and polymerase η foci formation. After 3 h of incubation, cells were counted in a Burker chamber and mixed in the same amounts to avoid the over-representation of any type of cell line. The fixation step was carried out using 3% PFA solution for 10 min. The cells were suspended and dropped to poly-L-lysine-covered slides. After the fixation, the sample was washed with PBS, followed by the staining of the nuclei with 0.5 μg/ml DAPI solution in PBS and then washing with MQ. The samples were kept in a humidity chamber until microscopic analysis to prevent drying.

**Direct amplification and sequencing of shDNA fragments from human cells**. Cells were captured in 5 μl catapult buffer, 150 cells for each phenotype (0.1 mM EDTA, 1 mM Tris pH 8, 0.5% Igepal). After capture, we added 0.5 μl Proteinase K (1 mg/ml) to the samples and incubated them at 60 °C for 20 min, followed by 3 min at 98 °C. Next, a two-step amplification reaction was carried out in 20 μl volume. In the first PCR, we used 10 μM shDNA-specific primer pair with a universal tag sequence, 1 × PCR buffer, 2.0 mM MgCl₂, 2.5 mM dNTPs, and 1 unit of AmpliTaq Gold DNA Polymerase (Thermo Fisher, Cat. No. 8080241). Thermal cycler conditions were: 95 °C for 2 min, 25 cycles of 95 °C for 15 s, 60 °C for 15 s, 72 °C for 30 s, and finally 2 min at 72 °C. In the second PCR, 1 μl from the first amplification reaction was used as a template with primers complementary to the universal tag sequence. The 5' end of the primers consisted of Illumina specific adaptor sequences. Other PCR conditions were the same as the first ones performing 30 PCR cycles this time. PCR reactions were run on a 2% agarose gel for amplification quality control. Successfully amplified samples were quantified using the qPCR-based quantification method (Kapa Biosystems, Cat. No. KK4854) on LightCycler480 qPCR (Roche, Indianapolis, IN). Finally, Illumina sequencing was carried out on the Illumina MiSeq system with Standard Flow Cell v2 (Illumina, Cat. No. MS-102-2002), following the manufacturer's instructions. Sequencing data were analysed using proprietary NGSeXplorer bioinformatics software. Sequencing reads were mapped to a reference sequence that contained all the 10 shDNA-specific sequences. Read counts were measured at each shDNA sequence. These data were then used to calculate the shDNA patterns for the two phenotypic groups (foci-forming and homogeneous cells).

**Limiting factors and sources of error**. Our approach is capable of isolating from several hundred to over a thousand cells per day. However, the throughput of CAMI is still limited by several bottlenecks. In the imaging and set-up stage, the main bottleneck is the microscope set-up (locating the landmark, configuring the microscope settings, finding focus, and selecting the region of interest). This comes at a fixed time cost per sample slide (see Supplementary Note 3 for timings). In the software analysis stage, the main bottleneck is processing enough images to find the desired number of cells for isolation. Rare phenotypes require searching through more images to find interesting cells to isolate. In the isolation stage, the main bottleneck is the laser microdissection, which takes ~10 s per cell (catapulting and stage movement are substantially quicker than cutting). Depending on the experimental parameters (rarity of the phenotype, number of desired cells, etc), the most costly bottleneck changes (Supplementary Note 3). When collecting relatively few cells (less than 100) of a common phenotype, the imaging and its set-up is the limiting factor. For very rare cell types that require the software to process thousands of images to find isolation candidates, the analysis software is the limiting factor (although this can be mitigated using distributed computing). When a large number of cells are desired (more than 1000), laser-cutting is the limiting factor.

Because CAMI relies on a diverse set of complex technologies, there exist several potential sources of error. It is difficult to mount a slide in perfect alignment with the stage for different microscopes, so there is potential for angular misalignment between microscope coordinates (multiple landmarks can mitigate this). When setting up the microscope, the user must select the appropriate areas of the sample to image or they may have difficulty finding isolation candidates. When imaging, problems with focus and artifacts in the image can cause errors in cell segmentation. The segmentation software itself is prone to errors, and the machine-learning predictions are imperfect. Errors in cell isolation contours may be caused by imperfect registration between microscope coordinates, which can result in poorly cut cells. In the laser microdissection step, losing focus or choosing the wrong cutting speed may result in failure to properly cut the cell (slow cutting can burn the cell, fast cutting can cause problems ejecting). Finally, the catapulting laser must be correctly calibrated or cells may be lost in the collection step.

**Data availability**. The authors declare that the data supporting the findings of this study are available within the paper and its supplementary information files.

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

## Acknowledgements

A.B., C.B., T.B. and P.H. acknowledge the Hungarian National Brain Research Program (MTA-SE-NAP B-BIOMAG). P.H. and L.P. acknowledge support from the Finnish TEKES FiDiPro Fellow Grant 40294/13. N.F. and G.T. were supported by the National Research, Development and Innovation Office of Hungary (VKSZ-14-1-2015-0155), the Hungarian Academy of Sciences and the ERC INTERIMPACT project. P.H. and I.N. were supported by the János Bolyai Research Scholarship of the Hungarian Academy of Sciences. L.H. and P.H. acknowledge the European Union and the European Regional Development Funds (GINOP-2.3.2-15-2016-00006, GINOP-2.3.2-15-2016-00020).

## Author contributions

P.H. and K.S. conceived the project. P.H. led the project. G.T. and L.H. co-supervised the RNA and DNA studies, respectively. C.B., A.B., F.S. and L.H. executed the pipeline and performed single-cell isolation. C.M., A.B., L.P., T.B., A.S. and P.H. wrote the software components. C.B., N.F. and L.G.P. performed dPCR analysis. C.B., B.B. and I.N. performed transcriptome analyses. L.H., M.E. and L.H. performed DNA analysis. K.S., C.B., L.H., C.M., A.B., I.N., L.P., L.H., G.T. and P.H. wrote the manuscript. All authors read and approved the final manuscript.
