## [Peer Review File · Nature Communications]

Reviewers' comments:

Reviewer #1 (Remarks to the Author):

The paper describes a workflow for computer assisted isolation of individual cells using image analysis, machine learning, and laser-capture microdissection.

The system, called CAMI, is demonstrated on three different experiments:

1. Correlating protein expression levels with mRNA quantification on a per cell level in rat cerebral cortex.
2. Morphology-based selection and isolation of pyramidal cells from rat cerebral cortex followed by gene expression profiling.
3. Isolation of cells with 'interesting phenotypes' in a mixed population of stable silenced cell lines exposed to DNA damage. Captured cells were sequenced using NGS, and regulators were identified. This is a nice suggestion for an alternative to RNAi-based screening.

The paper is very well written, the structure is easy to follow, and results are clearly presented.

The novelty lies in connecting automated image analysis and single-cell feature-based classification with automation of laser-capture microdissection (LCM). Each of the two has been thoroughly used and described before, but combining them in a single workflow seems novel.

However, my main concerns are the limitations of this combined system and sources of error of the presented approach.

It is stated that 'Computer-driven automation increases throughput over manual techniques by orders of magnitude (a few thousand cells per day with CAMI compared to ten with patch clamp harvesting) and microscopy-based isolation boasts several advantages over conventional high throughput isolation techniques.'

However, the authors do not use 'a few thousand cells' in any of their experiments.

In Exp 3, 150+150 cells are isolated.

In Exp 2, 50 cells are isolated.

In Exp 1, no cell number is given. The Supplementary says >50 of each cell type was used for for training, but the results in figure 2e only shows measurements from 13 cells. If more than 50 cells were manually selected and used for training, but only 13 cells make it to the final step of the experiment, is there really a gain to automate the selection with CAMI? This raises a number of questions: What are the limiting factors? What are the largest sources of error? How important is the visual confirmation? How many cells rejected at this stage?

It is noted that a version of CellProfiler that is more than 10 years old was used in the experiments, and that a plugin that can handle overlapping cells (written by the authors) was added to the CellProfiler pipeline.

In the end, overlapping cells seem to be discarded as unreliable in the selection and microdissection step (a conclusion drawn by me based on the illustrations, but not discussed in detail in the paper). This raises two questions: Why bother segmenting overlapping cells when isolated cells are preferred to avoid contamination at microdissection?

Why use a 10 year old version of CellProfiler when there are several revisions with improved methods for segmentation as well as illumination correction, and the version presented here is more or less abandoned? Also, CellProfiler Analyst should be referenced and compared to the machine learning interface described here.

Coordinates of cells in images from the two microscope systems are aligned based on a single etched landmark. Differences in resolution between source and target (HT system and

microdissection microscope) should be discussed. Is there a need for multiple landmarks to confirm registration; this must be crucial for the correct dissection of cells: Was this evaluated?

The CAMIO user manual says:

'The main functionality of the CAMIO is to move contour data that was extracted from images of the high-content screening microscope'.

Does this mean that CAMIO is mainly for contour editing? Is this due to poor alignment or poor segmentation?

What is the time gain between automated/manual correction and purely manual selection? The authors should clarify the manual steps in the training and selection procedure. Would it be possible to image the samples and do image analysis based on data collected using the microdissection microscope to avoid alignment errors?

Minor comments:

Is there an explanation for the fluorescent artefacts in Fig 2 c?

Fig 2 g has very poor image contrast; please increase for readability.

It is stated that the authors have competing financial interests. Is this a typo, or is there an explanation missing?

The manual for CAMIO contains several spelling errors, please run through a spell check.

Reviewer #2 (Remarks to the Author):

*** Overview

This paper presents an image-based cell isolation method, which is supposed to be high throughput, non-disruptive and cost effective. The pipeline comprises imaging -> illumination correction (using Cidre) -> cell segmentation (using the authors publication in Scientific Reports 2016) -> machine learning to predict cellular phenotypes (based on manual labeling and cell profiler) -> phenotype-dependent isolation -> (single cell) molecular analysis.

An automated pipeline to image, classify, and process single cells using the morphological and micro-environmental information of cells is an important and original contribution. In the present manuscript however, I have a number of issues with the motivation of the method, the three use cases, and the presentation of the approach.

*** Major issues

- Scope. The authors mention the importance of the microenvironment of the cells. However, they as far as I can see, the microenvironment is not taken into account when cells are classified with the machine learning software (although this is claimed on page 3 'cell types within the same microenvironment') and then selected for further processing. Moreover, if the method is 'disturbing the microenvironment' (page 2) or not is irrelevant for the cells that are further processed. A better motivation is needed here.

- Applications. Also the three applications are not well motivated from my perspective: In application 1, only few cells (13?) are analysed, which requires for sure no automatic identification and processing of cells. The same is true for the 50 cells in application 2. Application 3 might be more relevant, but I could not follow the description of the experiment and the goals fully. I suggest to present one strong application that showcases the usefulness of all aspects of the method (e.g. the 'thousand cells

per day' as mentioned in the discussion), rather than three rather weak examples.

- Evaluation. The machine learning prediction has no quantitative evaluation, neither is the amount of manual annotation they require discussed. Moreover, how much manual intervention is needed to correct the automatic outputs? These details are important criteria to judge on how high-throughput the method can be and how useful it is for other researchers.
- The presentation of the results can be improved (see below)

*** Minor issues

- Page 2, line 47: Please describe why 'isolation methods' are important. And: Is this really the appropriate term for FACS?
- LCM is not used consistently
- Page 2, line 54: 'interpreting analysis data' is a bit strange as an expression? Do you mean processed data?
- Page 3, line 111: Are the 'two biological replicates' from the same tissue sample? Please comment on: Do you have to retrain your classifier on each replicate? Or can you use the previously trained classifier?
- Line 123: How can silenced genes be identified via sequencing? A non-detect doesn't mean the gene is silenced.
- More relevant literature on techniques with similar possibilities like MALDI and imaging mass cytometry (e.g. Charlotte Giesen, H. A. O. Wang, Denis Schapiro, Nevena Zivanovic, Andrea Jacobs, Bodo Hattendorf, Peter J Schüffler, Daniel Grolimund, Joachim M Buhmann, Simone Brandt, Zsuzsanna Varga, Peter J Wild, Detlef Günther & Bernd Bodenmiller, Nature Methods 2014) have to be cited and discussed.

**** Figure 1:

- b: Why are there three slides on top of each other?
- c: What does this figure tell me? You refer to 'Cell' features on the right?
- g is not part of the pipeline. Or is it? Please specify.

**** Figure 2:

- The colors in b are a bad choice, cells are hard to discriminate. Use dotted/dashed lines instead.
- d: Are these cells the same as in b? They should be. The panel next to it: What does it show? Why is the Hoechst staining missing in the left examples?
- Instead of e, show the correlation between protein and mRNA.

Reviewer #3 (Remarks to the Author):

The manuscript by Brasko et al., presents a technique designed to enable high throughput phenotypic identification and laser microcapture of cells within fixed tissue sections or cultured cells. The technique is timely and of potential broad relevance given the current interest in high throughput single cell molecular analyses across many fields of study. However, I think that additional data (summarized below) is needed in order to be able to understand if this technique represents a significant advance that can be broadly useful to the community. My specific comments to the authors are as follows:

1. The authors make the argument that one of the advantages of the technique described in the paper is that it facilitates high throughput capture of single cells in situ without the disruption induced by cell dissociation and subsequent FACS analysis used in many studies, allowing for

preservation of the spatial context of the captured cells. However, the proof of principle experiments presented focus largely on pooled cell samples. For the single cell analysis, the authors use CAMI to extract nNOS-positive cells from the rat cortex and then use digital PCR to assess the expression of nNOS and a housekeeping gene. This provides limited resolution information on the applicability of this technique to single cell analyses, as it is difficult to discern how the single cell RNA captured with this technique compares to the RNA captured in studies that have reported good success with single cell RNA-seq using whole cell dissociation methods (such as Tasic et al., 2016, Nature Neuroscience). Ideally, the authors would extract single cells and perform RNA-sequencing to analyze the whole single cell transcriptomes, similar to the experiment they present with a pool of 50 presumed pyramidal neurons. The results could then be compared in a more direct fashion with the data from published studies of single cell neuronal transcriptomes. This would lend strong support to the idea that this technique is broadly useful for single cell transcriptomic analyses.

2. For the experiments comparing nNOS labeled interneurons with presumed pyramidal neurons, how does the cell isolation tool that is employed differentiate between DAPI-positive pyramidal neuronal nuclei and nNOS-negative interneurons and non-neuronal cells? Is the selection based solely on nuclear size or shape? It is difficult to assess the fidelity of this method for extracting cells of a particular phenotype given that only 4 nNOS-negative cells are shown and one of them appears to have low expression of nNOS.

3. For the experiment in which 50 presumed pyramidal neurons are pooled for sequencing, it would be useful to provide more information about the sequencing results so that the reader can better compare the data to similar studies. For example, it would be useful to summarize the alignment statistics (%mRNA, gDNA, etc.) and state the number of genes detected in each sample. Without these values it is difficult to assess the quality of the sequencing data. Can the authors also explain the rationale for using a total RNA kit for preparing their samples rather than focusing on mRNA? I would also like to see an example of a cDNA profile from one of these samples (e.g. Bioanalyzer trace or similar) so that cDNA library quality can be visually assessed by the reader. Furthermore, a comparison of the pooled samples to a control sample, such as a similar quantity of control RNA (i.e. total cortical RNA or similar), would be helpful. This would allow for a comparison of the microdissected specimens to a standard which would aid in interpretation of the quality and utility of the sequencing data obtained from fixed, microdissected specimens.

4. It seems as though the segmentation used targets cutting around the nucleus of a selected cell implying that cytoplasmic RNA is largely lost from the sampled cells. Does this limit the utility of the technique to larger cells that may have higher nuclear RNA content? Is it possible to use this technique to target cells with small nuclei, such as glial cells, and successfully obtain sufficient RNA for downstream analyses?

5. How are cells bisected by the sectioning process handled by the software (i.e. cell caps, partial cells)? Does the cell selection software used avoid these partial cells? Presumably, partial cells would contain less RNA than cells contained entirely within the section and would produce less useful downstream information for many applications, such as whole transcriptome analysis.

Response to Reviewer feedback

In the following, the comments from the reviewers have been enumerated and *italicized*. Our response immediately follows each comment. The first comment from Reviewer 1 is denoted C1.1, and our response is denoted R1.1.

Response to Reviewer 1

C1.1 *It is stated that 'Computer-driven automation increases throughput over manual techniques by orders of magnitude (a few thousand cells per day with CAMI compared to ten with patch clamp harvesting) and microscopy-based isolation boasts several advantages over conventional high throughput isolation techniques.' However, the authors do not use 'a few thousand cells' in any of their experiments... [In experiment 1] more than 50 cells were manually selected and used for training, but only 13 cells make it to the final step of the experiment, is there really a gain to automate the selection with CAMI?*

R1.1 One of the main advantages of the CAMI system is the intelligent and objective selection of samples through image analysis and machine learning. Our approach allows us to inspect many cells and to carefully select the most promising ones for analysis. We are able to inspect orders of magnitude more cells than we ultimately isolate and analyze.

However, the reviewer correctly pointed out that the experiments provided in our original manuscript do not adequately demonstrate the high-throughput capabilities of the CAMI system. Our updated manuscript addresses this by 1) increasing the number of isolated cells in experiment 2 and 2) including a new Supplementary Note 3, which reports the time costs for isolating cells. The main text and Figure 2 have been updated to reflect these changes.

C1.2 *This raises a number of questions: What are the limiting factors? What are the largest sources of error? How important is the visual confirmation? How many cells rejected at this [visual confirmation] stage?*

We have added a section **limiting factors and sources of error** at the end of the online methods describing in detail the limiting factors of the method as well as the sources of error. The visual confirmation before contour generation was only used to discard cells which possibly have close neighbours, and this step was to avoid contamination.

C1.3 *Why use a 10 year old version of CellProfiler when there are several revisions with improved methods for segmentation as well as illumination correction, and the version presented here is more or less abandoned? Also, CellProfiler Analyst should be referenced and compared to the machine learning interface described here.*

R1.3 Based on the reviewer's comments, we have taken a significant step to rewrite a large portion of our code in Python and update it to be compatible with the current version of CellProfiler. This includes the module for detecting overlapping cells, the secondary object detection module, and the intensity, texture, and shape measurement module (**Supplementary Software 1**). We also updated the pipelines to be

compatible with the current version of CellProfiler (**Supplementary Software 2**). Our instructions in **Supplementary Note 2** have been updated to reflect the changes in the software.

The reviewer made an important comment highlighting the need to compare to other software such as CellProfiler Analyst. The main focus of this brief communication is a unique combination of technologies that allow high-throughput, targeted, non-disruptive single-cell isolation. CellProfiler Analyst is only one of many software packages designed to perform machine learning driven phenotypic analysis. If we expand our manuscript to compare to CellProfiler Analyst, to be fair and comprehensive, we should also compare to other open-source software packages such as Enhanced CellClassifier, WND-CHARM, or cellXpress. We chose to use Advanced Cell Classifier (ACC 2.0) because it has a large library of machine learning algorithms to select from, and it has several features that make annotation quick and easy: active learning, a phenotype finder, and a similar cell search tool¹. Coincidentally, several of our authors have been invited to send a review article to *Cell Systems* comparing these software tools. We are happy to share our experience and have attached a table we prepared for that manuscript, which compares open source phenotypic analysis tools.

1. Piccinini, Filippo, et al. "Advanced Cell Classifier: User-Friendly Machine-Learning-Based Software for Discovering Phenotypes in High-Content Imaging Data." *Cell Systems* (2017).

	WND-CHARM (Orlov et al., 2008)	CP-CHARM (Uhlmann et al., 2016)	CellCognition (Held et al., 2010)	Ilastik (Sommer et al., 2011)	cellXpress (Laksameethanasan et al., 2013)	HCS-Analyzer (Ogier et al., 2012)	CellClassifier (Raimo et al., 2009)	Enhanced CellClassifier (Müsselwitz et al., 2010)	CellProfiler Analyst v2.0 (Dae et al., 2016)	ACC v2.0
DOCUMENTATION										
Handbook	○	○	○	●	●	●	●	●	●	●
Website	●	○	●	●	●	●	●	○	●	●
Video tutorial	○	○	○	○	○	●	○	○	●	●
Open source code	●	●	●	●	○	●	●	●	●	●
Test dataset	○	●	●	●	●	●	○	●	●	●
USABILITY										
No programming experience required	○	●	●	●	○	●	●	●	●	●
User friendly GUI	○	●	○	○	○	●	●	●	●	●
Intuitive visualization settings	○	●	●	●	●	●	●	●	●	●
Does not require commercial licence	●	Matlab	●	●	●	●	Matlab	Matlab	●	Matlab/●
FUNCTIONALITY										
Plate/image selection	○	○	●	●	●	○	○	○	●	●
Time-lapse analysis	○	○	●	●	○	○	○	○	○	○
3D analysis	○	○	○	●	○	○	○	○	○	○
Image segmentation	○	○	●	●	●	○	○	○	○	○
Feature extraction	●	●	●	●	●	○	○	○	○	○
Supervised classification	○	○	●	●	○	○	●	●	●	●
Unsupervised classification	●	●	○	○	○	●	○	○	○	●
Active learning	○	○	○	○	○	○	○	●	●	●
Similarity search	○	○	○	○	○	○	○	○	●	●
Outlier discovery	○	○	○	○	○	○	○	○	○	●
OUTPUT										
Visual cell classification	○	○	●	●	○	○	○	●	●	●
Feature-based statistics	○	○	○	○	○	●	○	○	○	●
Class-based statistics	●	●	●	●	○	●	●	●	●	●
Plate-based statistics	○	○	○	○	○	●	●	●	●	●

Signs and abbreviations: ●, available; ○, not available; GUI: Graphical User Interface

C1.4 *In the end, overlapping cells seem to be discarded as unreliable in the selection and microdissection step. This raises two questions: Why bother segmenting overlapping cells when isolated cells are preferred to avoid contamination at microdissection?*

R1.4 Without the wavelet transform approach we developed to segment overlapping cells, we do not have a reliable way to identify overlapping cells at all. Classification using CellProfiler extracted features is partially successful, but fails to handle difficult cases well. Our solution was to recognize overlapping cells by developing a method to reliably segment them. An added benefit of the approach we developed is that if the users want to perform a phenotypic analysis of the cells (not just cut them), our approach can provide more reliable segmentations and thus a more precise analysis.

To make explicitly clear for the reader, we have added a sentence to the online methods:

“This method allowed us to reliably identify cells with overlapping nuclei, which are typically discarded from molecular analysis.”

C1.5 *Coordinates of cells in images from the two microscope systems are aligned based on a single etched landmark. Differences in resolution between source and target (HT system and microdissection microscope) should be discussed. Is there a need for multiple landmarks to confirm registration; this must be crucial for the correct dissection of cells: Was this evaluated?*

R1.5 We have added **Supplementary Note 1**, which contains a description of the laser-etched landmarks, an evaluation of the accuracy of registration using a single landmark, and a comparison to registration with multiple landmarks. Our findings indicate that if the working area is near the landmark, a single landmark has sufficiently accurate registration (1.7 μm mean registration error). Registration using two or three landmarks reduces registration error for applications where it is necessary.

C1.6 *The CAMIO user manual says: 'The main functionality of the CAMIO is to move contour data that was extracted from images of the high-content screening microscope'. Does this mean that CAMIO is mainly for contour editing? Is this due to poor alignment or poor segmentation? What is the time gain between automated/manual correction and purely manual selection? The authors should clarify the manual steps in the training and selection procedure. Would it be possible to image the samples and do image analysis based on data collected using the microdissection microscope to avoid alignment errors?*

R1.6 The main purpose of CAMIO is not contour editing. It is designed to review the cells automatically selected for extraction (or to manually select cells) and to transfer their contours between microscopes. The vast majority of automatically generated contours do not require any corrections. In our experiments we did not manually edit any of the cutting contours. However, we recognize that segmentation errors may occur in the contour of a rare or interesting cell, and for this reason we have included the ability to edit contours in CAMIO.

The time gained by accepting automatically selected cells is roughly between one and five seconds per cell, using our interface. If the user wishes to manually select cells but does not correct the contour, the time is less. If the user wishes to correct a contour, the user draws a new contour and we discard the automatically generated contour. The process takes about 2 to 10 seconds.

The manual step in the training process involves annotating cells with class labels using Advanced Cell Classifier. In general, it is necessary to train a classifier for each assay, and the number of annotations

required change with the difficulty of the classification task. In our experiments, we found that $n=78$ training annotations were sufficient for the neuron experiment and $n=41$ annotations were sufficient for the foci-forming experiment. Each annotation takes about 3-5 seconds (Supplementary Figure 3). In Supplementary Note 3, we have included a table summarizing the timing of every step in the proposed workflow, including the manual ones.

Finally, yes it is possible to use the same microscope for screening and isolation, and this would increase the precision of the cutting process. Unfortunately, it is our experience that laser micro-capturing microscopes have serious limitations when it comes to automation, focus detection, and the quality of fluorescence imaging. In practice, quick, high-quality and high-resolution imaging (such as the Operetta) are necessary for high throughput automatic cell analysis and selection. We remark that we have had discussions with microscope manufacturers about building isolation and acquisition systems together, and we hope that this manuscript will help trigger their interest.

C1.7 *Is there an explanation for the fluorescent artefacts in Fig 2 c?*

R1.7

It's not clear exactly what the reviewer is referring to. If he/she is talking about bright artifacts near the nuclei in areas distal from the cutting edges (arrow *a*), these are due to crystallization of the fluorophores caused by tissue aridity. The tissue sample presented in Figure 2 was first imaged with an automated screening microscope, then inserted to the isolation instrument. For the purpose of visualizing isolated cells, it was then inserted in the automated microscope again for imaging. This extended time allowed the sample to dry and crystallize, which can be observed in Fig 2. The extended imaging time was necessary to generate a high quality image for the figure. It is not normally performed, so handling time and exposure is shorter and this is typically not an issue.

If the reviewer is referring to the yellow haze (arrow *b*), this is cutting flash (also called laser cutting ejecta). During the cutting phase, laser microdissection causes tissue damage which results in fluorescent artifacts [Laser material processing. DOI 10.1007/978-1-84996-062-5]. We measured the extent of the damaged area (smaller than $2.5\ \mu\text{m}$) and we typically extend the cutting contours of the cells by $3\ \mu\text{m}$ to compensate for the cutting flash.

C1.8 *Fig 2 g has very poor image contrast; please increase for readability.*

R1.8

We have increased the contrast of Figure 2g as the reviewer requested. We have also changed the contours to match the other panels. Extracted cells have dashed-line contours. Colors have been changed to increase visibility.

C1.9 *It is stated that the authors have competing financial interests. Is this a typo, or is there an explanation missing?*

R1.9

An explanation is omitted by mistake. The manuscript now includes the following explanation:

P.H. is the founder and shareholder of Single-cell technologies Ltd.; L.P. is the founder and shareholder of Avidin Ltd. which provided SingleCellProtect reagent; B.B. and I.N. had consulting positions at SeqOmics Biotechnology Ltd. at the time the study was conceived. Single-cell technologies Ltd., Avidin Ltd. and Seqomics Biotechnology Ltd. were not directly involved in the design and execution of the experiments or in the writing of the manuscript. This does not alter the author's adherence to all the Nature policies on sharing data and materials.

C1.10 *The manual for CAMIO contains several spelling errors, please run through a spell check.*

R1.10

We have proofread the CAMIO manual and corrected spelling and grammar errors.

Response to Reviewer 2

C2.1 *The authors mention the importance of the microenvironment of the cells. However, they as far as I can see, the microenvironment is not taken into account when cells are classified with the machine learning software (although this is claimed on page 3 'cell types within the same microenvironment') and then selected for further processing. Moreover, if the method is 'disturbing the microenvironment' (page 2) or not is irrelevant for the cells that are further processed. A better motivation is needed here.*

R2.1

The sentence on page 3 reads "The dPCR results confirm that single nNOS-expressing neurons were reliably separated from other cell types within the same microenvironment based only on immunohistochemistry and phenotypic image analysis." It was not our intention to claim that the machine learning explicitly accounts for the microenvironment. Our claim is that we were able to isolate cells of a specific type that were in close proximity to other cell types within the same tissue sample. We use image analysis and machine learning to accomplish this, but we do not explicitly model the microenvironment. We have adjusted this sentence in the manuscript to clarify.

We agree with the reviewer there are many cases where disturbing the microenvironment may be irrelevant to further processing. However, it cannot be denied that the location of a cell and its microenvironment influences a cell in many ways. One important advantage of our technique is that it retains exact knowledge of the original location in the tissue of every cell we isolate, and collects a high resolution image of the environment. This opens the possibility for new types of experiments that correlate molecular analyses to a cell's original location in the tissue. For example, our technique makes it possible to compare molecular profiles of the same type of cells from different locations within the same tissue sample. Our first experiment is a proof-of-concept of this idea, in which we isolated cells from specific layers (L2 and L3) of the somatosensory cortex. Some knowledge of the microenvironment can be inferred using the location of the cell and image analysis of visual phenotypes. For example, it is possible to calculate the number of adjacent glial cells to the isolated neuron.

C2.2 *The three applications are not well motivated from my perspective: In application 1, only few cells (13?) are analysed, which requires for sure no automatic identification and processing of cells. The same is true for the 50 cells in application 2. Application 3 might be more relevant, but I could not follow the description of the experiment and the goals fully. I suggest to present one strong application that showcases the usefulness of all aspects of the method (e.g. the 'thousand cells per day' as mentioned in the discussion), rather than three rather weak examples.*

R2.2

Our goal in presenting three applications was to show the variety of applications enabled by our approach and its other advantages. After careful design considerations, we choose examples related to cancer and brain research, using cell lines and tissue samples, with RNA and DNA studies. Our choices show that our approach can be applied to fixed cells, that it retains location information of the cells does not lyse the cells, and studies can be performed on individual cells or pooled cells. We hope that through these demonstrations that we reach a broad audience of researchers studying single cells and convince them that our approach can fit their applications.

We acknowledge the reviewer's point that our manuscript did not convincingly demonstrate the high throughput nature of the technology. Therefore, we have performed an additional series of experiments on hundreds of single cells to demonstrate scalability of the method. In particular, have:

1. collected 300 pyramidal cells and sequenced the RNA obtaining ~3 million raw reads (designated 300x nNOS- Replicate 3)
2. collected 50 astrocytes and sequenced the RNA obtaining ~4 million raw reads (designated 50x Astrocytes)
3. increased the depth of sequencing of the two previously reported experiments shown in the first version of the manuscript (50 pyramidal cells; designated as 50x nNOS- Replicate 1 and Replicate 2) from ~2 million to >4 million raw reads

The results of the new experiments show that we are able to collect cells in a high throughput setting (the 300 cells were collected in a 3 hour period), and sequence the cells with good fidelity (**Supplementary Table 2**). The analysis in **Figure 2f** shows a high correlation between three nNOS- replicates of differing sizes from different animals, and negligible correlation between nNOS- cells and astrocytes.

In the revised version of the manuscript we have updated **Supplementary Table 2** so that top 100 genes are shown. These data show that sequencing the three replicates of pyramidal cells identified largely overlapping gene pools. Sequencing of the RNA derived from astrocytes identified a unique gene expression pattern.

Our updated manuscript contains a more clear explanation of application 3.

C2.3 *The machine learning prediction has no quantitative evaluation, neither is the amount of manual annotation they require discussed. Moreover, how much manual intervention is needed to correct the automatic outputs? These details are important criteria to judge on how high-throughput the method can be and how useful it is for other researchers.*

R2.3

We have added **Supplementary Figure 2** which contains confusion matrices and cross-validation accuracies for each experiment. 157 training samples were collected for the neuron experiment, and 165 samples were corrected for foci-forming experiment. The cross-validation accuracy was 91% and 95%, respectively. The most confident samples were isolated for each class, and there was no manual intervention.

There is a trade-off between the cost of collecting annotations and the accuracy of the classifier, which increases as more annotations are provided. The classifier learns to fit the data better with more annotations, and the cross-validation estimate of the accuracy becomes more robust. At the bottom of Supplementary Figure 2, a table illustrates how accuracy changes as more samples are annotated. For the neuron experiment, it is likely that somewhere around 70 annotations is sufficient. For the foci-forming experiment, the cross-validation accuracy stabilizes with less than 40 annotations.

A reference to this material has been included in the online methods.

C2.4 *Page 2, line 47: Please describe why 'isolation methods' are important. And: Is this really the appropriate term for FACS?*

R2.4

The purpose of the introductory paragraph of the article is to briefly motivate why isolation methods, and in particular single-cell isolation methods, are important. Our point may not have been clear because we

failed to introduce the term until the following paragraph on page 2. In our revised manuscript we have modified the text to emphasize that isolation methods are important for molecular analysis because cells must be isolated at the single-cell level and selected based on individual characteristics. We also stress that a disadvantage of previous techniques is the loss of environmental information derived from the localization of the cells.

Regarding the use of the term 'isolation method', we believe this to be a suitable term for the group of methods designed to target and isolate cells from tissue or medium, including FACS. Previous usages of the term can be found in the literature, such the excerpt from [1] (p. 226):

"Isolation method. Gene expression profiling of different leukocyte subsets recently demonstrated that FACS sorting is the recommended method for isolation of leukocyte subsets for gene expression studies since this method results in the purest subset populations and does not appear to induce a stress response. This is the best available isolation method at the moment..."

A Google search using the terms 'FACS "isolation method"' returns 64,200 results, including [2] which also uses the term referring to FACS. If there is a more appropriate term we are open to suggestions.

1. Cassetta, Luca, et al. "Isolation of Mouse and Human Tumor-Associated Macrophages." *Tumor Microenvironment*. Springer International Publishing, 2016. 211-229.
2. Beliakova- Bethell, Nadejda, et al. "The effect of cell subset isolation method on gene expression in leukocytes." *Cytometry Part A* 85.1 (2014): 94-104.

C2.5 *LCM is not used consistently*

R2.5

We have corrected the manuscript so that it consistently uses LCM throughout.

C2.6 *Page 2, line 54: 'interpreting analysis data' is a bit strange as an expression? Do you mean processed data?*

R2.6

We mean to say 'interpreting the results of genetic analysis'. We adjusted the text to clarify.

C2.7 *Page 3, line 111: Are the 'two biological replicates' from the same tissue sample? Please comment on: Do you have to retrain your classifier on each replicate? Or can you use the previously trained classifier?*

R2.7

The samples of the two biological replicates derived from two different rats. To keep the circumstances consistent in both cases, we collected the same cell type from the same cortical area in both animals (somatosensory cortex).

It is our experience that a classifier trained on one data set does not transfer to another reliably - there are too many sources of variation between biological replicates and data acquisition conditions. Our previous work on cell lines handled fully automatically (eg Wild et al. Plos Biology, Laurell et al. Cell) convinced us that, even in many highly controlled cases, classifiers do not generalize well. Therefore, also in this work we trained a separate classifier for each experiment/replicate.

C2.8 *Line 123: How can silenced genes be identified via sequencing? A non-detect doesn't mean the gene is silenced.*

R2.8

All the cells present in the used cell library carry a construct with a gene-specific or control shRNA sequence. Thus, the silenced gene can be identified for each cell. We have modified the text in the manuscript to clarify this point.

The clarification states:

“Once the cells are extracted, they can be sequenced to identify the silenced gene using universal primers flanking the specific silencing shRNA region present in each cell of the library.”

C2.9 *More relevant literature on techniques with similar possibilities like MALDI and imaging mass cytometry (e.g. Charlotte Giesen, H. A. O. Wang, Denis Schapiro, Nevena Zivanovic, Andrea Jacobs, Bodo Hattendorf, Peter J Schüffler, Daniel Grolimund, Joachim M Buhmann, Simone Brandt, Zsuzsanna Varga, Peter J Wild, Detlef Günther & Bernd Bodenmiller, Nature Methods 2014) have to be cited and discussed.*

R2.9

The reviewer points out that we did not make reference to single-cell mass spec methods. Although the cited paper is interesting, it misses the main purpose of our approach - that is to automatically select and isolate cells of interest - not to scan the whole sample. However, nowadays several automated targeted single-cell mass spec methods have been proposed that are capable of only measuring cells of interest, such as Fujii, Takashi, et al. "Direct metabolomics for plant cells by live single-cell mass spectrometry." *Nature protocols* 10.9 (2015): 1445-1456 which we have introduced as a new citation in the main text. This makes it orders of magnitude faster to make targeted measurements and more cost effective. We remark that these methodologies lack the possibility to intelligently pick phenotypes based on sophisticated machine learning decisions. We also remark that it is possible to integrate the CAMI technology presented in this paper into single-cell mass spec devices.

C2.10 *Figure 1b: Why are there three slides on top of each other? 1c: What does this figure tell me? You refer to 'Cell' features on the right? 1g is not part of the pipeline. Or is it? Please specify.*

R2.10

In several panels of Figure 1 we attempted to symbolize several aspects of the workflow through diagrams, some of which seem to be confusing. In Figure 1b, our goal was to represent the high throughput nature of the system by including a queue of slides to be imaged. In Figure 1c, our goal was to represent the image analysis process - the segmentation of the cells, the extraction of cell measurements, and the definition of cutting contours. Despite our best efforts, we could not think of better representations, so we have attempted to clarify these points by making the text in the figure more clear. Regarding panel 1g, single cell analysis is not part of the CAMI pipeline. However, we wanted to include this information so the reader can quickly understand from a single figure the possibilities of what can be done with captured cells. We have updated the figure caption to clarify these points and changed 'image features' to 'cell measurements' in panel 1c.

C2.11 *Figure 2: The colors in b are a bad choice, cells are hard to discriminate. Use dotted/dashed lines instead. 2d: Are these cells the same as in b? They should be. The panel next to it: What does it show? Why is the Hoechst staining missing in the left examples? Instead of e, show the correlation between protein and mRNA.*

R2.11

In the revised manuscript we have adjusted the colors and lines in Figure 2 to make them more clear. We have also included dashed lines to indicate cells that have been cut. Panel 2d shows cells that we collected. It is split in two parts: on the left side it is labeled nNOS+ interneurons, on the right it is labeled nNOS- cells. Four of the cells in panel 2d are the same cells isolated from the sample in panel b (two nNOS+ and two nNOS-). These cells are now marked with dashed lines. We have changed the text of the labels and added color bars to the figure so it is more clear to the reader. The Hoescht stain is not missing in any of the panels, but the intensity was weak on these panels. We discovered that on certain monitors it was very difficult to see. We slightly adjusted the contrast of the blue channel to improve visibility.

We realized that our description of the first experiment was misleading. We were not trying to directly compare protein levels to mRNA - we wanted to show that the machine learning selection of cells (which strongly relies on the immunofluorescent staining) corresponds to mRNA quantification. We have adjusted the manuscript to reflect this. "First, we check if immunofluorescent-labeled cells selected using machine learning in CAMI corresponded to mRNA quantification in individual neurons extracted from 10 μ m thick sections of rat cerebral cortex."

Response to Reviewer 3

C3.1 *The authors make the argument that one of the advantages of the technique described in the paper is that it facilitates high throughput capture of single cells in situ without the disruption induced by cell dissociation and subsequent FACS analysis used in many studies, allowing for preservation of the spatial context of the captured cells. However, the proof of principle experiments presented focus largely on pooled cell samples. For the single cell analysis, the authors use CAMI to extract nNOS-positive cells from the rat cortex and then use digital PCR to assess the expression of nNOS and a housekeeping gene. This provides limited resolution information on the applicability of this technique to single cell analyses, as it is difficult to discern how the single cell RNA captured with this technique compares to the RNA captured in studies that have reported good success with single cell RNA-seq using whole cell dissociation methods (such as Tasic et al., 2016, Nature Neuroscience). Ideally, the authors would extract single cells and perform RNA-sequencing to analyze the whole single cell transcriptomes, similar to the experiment they present with a pool of 50 presumed pyramidal neurons. The results could then be compared in a more direct fashion with the data from published studies of single cell neuronal transcriptomes. This would lend strong support to the idea that this technique is broadly useful for single cell transcriptomic analyses.*

R3.1

We respectfully disagree with the reviewer on whether the experiment presented for single cell digital PCR provides limited resolution information on single cell analysis. Single cell digital PCR is the gold standard used to validate single cell RNA-seq data, thus the resolution and accuracy of the example presented is superior relative to whole single cell transcriptomes. We believe that the insight of the reviewer is helpful for studies aiming primarily for cell classification where the quality of sequencing is crucial for single cell derived material like the Tasic et al. paper. However, the focus of our study is not to perform sequencing of similar quality on single cells. Our goal is to show methodologically that the material collected using CAMI is suitable for a wide range of applications based on single cell-, oligocellular- and multicell-derived material. We demonstrate that RNA content is preserved by CAMI from single cells down to the ability to detect minimal changes in mRNA copy numbers.

C3.2 *For the experiments comparing nNOS labeled interneurons with presumed pyramidal neurons, how does the cell isolation tool that is employed differentiate between DAPI-positive pyramidal neuronal nuclei and nNOS-negative interneurons and non-neuronal cells? Is the selection based solely on nuclear size or shape? It is difficult to assess the fidelity of this method for extracting cells of a particular phenotype given that only 4 nNOS-negative cells are shown and one of them appears to have low expression of nNOS.*

R3.2

We use machine learning algorithms to learn a nonlinear function to predict cellular phenotype through a learned nonlinear function of various cell measurements. In this case, a random forest classifier was used, which combined many features including size, shape, and texture features, among others.

C3.3 *For the experiment in which 50 presumed pyramidal neurons are pooled for sequencing, it would be useful to provide more information about the sequencing results so that the reader can better compare the data to similar studies. For example, it would be useful to summarize the alignment statistics (%mRNA, gDNA, etc.) and state the number of genes detected in each sample. Without these values it is difficult to assess the quality of the sequencing data.*

R3.3 In the new experimental setup we have used the REPLI-g WTA Single Cell Kit, and have sequenced all 4 samples on Ion Torrent PGM. We have added a new supplementary table summarizing the alignment statistics (Supplementary Table 3). We include a copy of the table below:

		sample			
		50x nNOS-R1	50x nNOS-R2	300x nNOS-R3	50x Astrocytes
total reads		5,304,896	4,127,917	2,961,059	4,048,433
total mapped		1,330,553	1,012,501	801,937	679,845
gene		835,320	617,179	440,108	517,000
intergenic		495,233	395,322	361,829	162,845
strand specificity	forward % of reads mapped	51.13	50.49	49.97	52.06
	reverse % of reads mapped	48.87	49.51	50.03	47.94

C3.4 *Can the authors also explain the rationale for using a total RNA kit for preparing their samples rather than focusing on mRNA? I would also like to see an example of a cDNA profile from one of these samples (e.g. Bioanalyzer trace or similar) so that cDNA library quality can be visually assessed by the reader.*

R3.4

Since we aimed to extract as much information as possible - including data on RNAs not coding proteins - we used total RNA as input. It turned out that even by using total RNA 99,60 – 99,97% of the detected biotypes fall within protein coding class.

Throughout the sequencing workflow all QC steps were performed on Agilent TapeStation. Below, we provide a copy of the electropherograms for all 4 cDNA pools sequenced:

Since the qualitative range of genomic DNA Screen Tape is between 10-100 ng/ μ l, we have first quantified the obtained cDNA using Qubit. The cDNA concentration of samples P, 2SN and A (all derived from 50 cells) fell within this concentration, hence they were used undiluted; the concentration of sample 300x nNOS- R3 (bottom left) derived from 300 cells had to be diluted in order to fit into the qualitative range.

C3.5 Furthermore, a comparison of the pooled samples to a control sample, such as a similar quantity of control RNA (i.e. total cortical RNA or similar), would be helpful. This would allow for a comparison of the microdissected specimens to a standard which would aid in interpretation of the quality and utility of the sequencing data obtained from fixed, microdissected specimens.

R3.5

As mentioned above, in addition to sequencing a third replicate of pyramidal cells (this time from 300 cells) we have opted to isolate and sequence the RNA derived from 50 astrocytes. Next, based on RPKM values we listed the top 100 genes from each of the 4 samples and compared the expression of these genes in a single table (see **Supplementary Table 2**). Since many of the top-100 genes in samples derived from pyramidal cells are overlapping, the table altogether contains 214 genes. The expression of 101 of these genes is exclusively detected in pyramidal cells (out of which the expression of 72 genes was detected in all three replicates; highlighted in red), while 57 genes are detected in astrocytes but in none of the 3 replicates of pyramidal cells (highlighted in green). In our opinion these data point toward the reproducibility of the experiments.

C3.6 It seems as though the segmentation used targets cutting around the nucleus of a selected cell implying that cytoplasmic RNA is largely lost from the sampled cells. Does this limit the utility of the technique to larger cells that may have higher nuclear RNA content? Is it possible to use this technique to

target cells with small nuclei, such as glial cells, and successfully obtain sufficient RNA for downstream analyses?

R3.6

The utility of the technique is limited for assays with high cell density. If the cells are not well separated or we cannot otherwise reliably segment the plasma membrane, we set the cutting contour to surround the nucleus with a 3 μm buffer to ensure that we collect material from only one cell. In this case, we can only recover partial cytoplasmic material. If the assay allows us to reliably segment cell membranes, we can extend the contours to fit just inside the cell membrane and recover most of the cytoplasmic RNA. For cells with small nuclei such as glial cells it should be safe to extend the cutting contour and collect sufficient cytoplasmic material for downstream analysis.

***C3.7** How are cells bisected by the sectioning process handled by the software (i.e. cell caps, partial cells)? Does the cell selection software used avoid these partial cells? Presumably, partial cells would contain less RNA than cells contained entirely within the section and would produce less useful downstream information for many applications, such as whole transcriptome analysis.*

R3.7

Our software is only provided a 2D image of the cell - if parts of the cell are missing from the sectioning process, it is potentially problematic as we do not have an explicit procedure to deal with this problem. However, in many cases, these issues will produce malformed cells or cells with small nuclei. Our software searches for certain sized nuclei - cells with too small or too large nuclei are not selected for isolation.

Reviewers' comments:

Reviewer #1 (Remarks to the Author):

The authors have made substantial revisions to the manuscript and added new data as well as clarifications of the text. One may still wish for a true large-scale experiment, but as the authors argue, the broad number of applications presented also has its merits. The authors have also provided thorough and valid responses to all review comments. I therefore recommend publication.

Reviewer #2 (Remarks to the Author):

Brasko and co-workers have clearly improved their manuscript by revising text and including more data. However, I still find there are a couple of major issues that have to be addressed properly to warrant publication in Nature Communications.

***** Evaluation of machine learning approach.

The authors now provide confusion matrices for two application and calculate accuracies for different numbers of samples.

- Please state the average time needed to annotate 157 or 165 samples with your software. Since any application will require its own training data set, this information is crucial for potential users.
- Accuracy drops from 93% to 91% for the nNOS cells. I suggest to calculate error bars for the accuracies (e.g. via bootstrapping) to illustrate when a stable plateau is reached. This is the crucial number of cells that have to be annotated I guess (supposed that the three classes are balanced).
- Is 'Trash' the same class as 'Discard' in Fig. 2?
- I suggest to use the same color code as in Fig. 2

***** Presentation

Fig 2 has a number of issues:

- Is 'Discard' in 2b a separate class that is used for training and prediction? For me, these 'Discard' nuclei look rather nice, at least I can see no 'artifacts' as stated in the figure caption. Pls detail why and how they are kicked out.
- Fig 2c does not show 'the same region' as 2b. It would be nice though.

***** Explanation

From my perspective, the third experiment is still not properly described:

- 10 different cell lines expressing 10 different shRNAs are mixed and tested for homogeneous vs. foci forming polymerase distribution. Is DNA or RNA of these cell lines sequenced at the end ('Read count ratio')?
- How do the authors discriminate between the 4 classes shown in suppl fig 1?
- Does 'Trash' and 'No GFP' in suppl fig 1 correspond to 'Discard' in fig 2g?
- I suggest to order the bars in Fig 2i.

***** Statistical analysis

Fig. 2e is supposed to show that selected cells differ in the expression of nNOS. I have a number of issues with this:

- A statistical test should be applied to prove that nNOS expression differs significantly while S18 does not.
- Fig 2d shows 6 nNOS negative cells. Only 4 nNOS neg cells are shown in Fig. 2e. Why only that few? Again, to promote the high-throughput potential of the method, all isolated cells should be shown.
- Logarithmic y axis labels are not necessary, neither for the few transcripts of nNOS, neither for the homogenous many transcripts of S18.
- The depicted micro tubes do not convey any information in Fig 2e and 2i, and they are misleading in Fig. 2f

Reviewer #3 (Remarks to the Author):

The manuscript by Brasko et al. details an imaging-based method for single cell isolation. While I believe that the single cell PCR and pooled cell sequencing data presented in the manuscript is sufficient for a brief proof of principle study, I have a few reservations about the study that are summarized below.

1. I agree with the authors that single cell PCR is a fine method for assaying gene expression on a gene by gene basis. My rationale for suggesting including further single cell sequencing data is that it provides a more global view of the transcriptome, which allows for assessment of measures such as gene dropout and library complexity that can be reflective of the quality of single cell libraries and can be valuable for assessing the utility of a given method for downstream applications. Perhaps this is beyond the scope of the study, but to convincingly demonstrate that this laser capture based method is a viable alternative to FACS isolation of cells of interest, I would want to see a more in-depth study of a larger population of single cells.
2. The authors use their single cell PCR data to demonstrate that RNA quality is relatively preserved after fixation, immunohistochemistry, and cell capture. Again, I find that this gives a limited view of the overall RNA quality within the captured cells because transcripts can vary in their rate of degradation and only a few genes were assayed in the present study. It would be helpful to see additional data supporting preservation of high quality RNA within the samples. An estimate of RIN from a small population of captured cells would be helpful. It does appear that there is some evidence of degraded RNA in the cDNA profiles provided as illustrated by the presence of low molecular weight products in the traces.
3. The authors state that they used a total RNA kit to prepare their pooled cell samples for sequencing because they wanted to extract as much information as possible, but I think that this has limited the amount of information that can be mined from their sequencing data. Without depletion of ribosomal RNA or use of an oligodT based method, the bulk of RNA captured and sequenced will be ribosomal. From the mapping statistics provided, it seems that the % of mapped reads is very low (~25% for neuronal samples and ~17% for astrocyte sample) which likely reflects most of the reads being occupied by sequencing of rRNA, but could also be influenced by sample quality, sequencing quality, and library complexity. Can the authors provide a value for the % of reads mapping to rRNA? The authors then provide a table of the top 100 expressed genes across their different replicates. Can they also provide a summary of the total number of genes detected? It looks like maybe only a few hundred genes were detected given that the list of top expressers includes many genes with low (<5) RPKM values. The authors state that even though they used total RNA for their sequencing experiments, they still mostly detected protein coding genes but the list of top expressing genes includes many uncharacterized and non-coding

transcripts. Where does the "99,60 – 99,97% of the detected biotypes fall within protein coding class" statement derive from? For example, for the 50x astrocyte sample, many of the genes expressed are U1 and U6 small nuclear spliceosomal RNAs. Also, it is interesting that many of the common marker genes for pyramidal neurons (SLC17A7) and astrocytes (GFAP, AQP4) are not among the highest expressed genes in the pooled samples. Are these detected at all? One would be hard pressed to guess what type of cells were sequenced given the gene list provided. Is there any expression of genes that do not correspond to the cell class captured in the pooled samples – i.e. expression of MBP or MOG in the neuronal or astrocyte samples? Is there any evidence of contamination from other cell types incidentally captured by the sampling technique?

Detailed response to reviewers

In the following, the comments from the reviewers have been enumerated. Our response immediately follows each comment. The first comment from Reviewer #2 is denoted R2.1, and our response is denoted A2.1. We highlighted changes in the main text with **yellow**.

Reviewer #2

R2.1

Evaluation of machine learning approach.

The authors now provide confusion matrices for two application and calculate accuracies for different numbers of samples.

- *Please state the average time needed to annotate 157 or 165 samples with your software. Since any application will require its own training data set, this information is crucial for potential users.*
- *Accuracy drops from 93% to 91% for the nNOS cells. I suggest to calculate error bars for the accuracies (e.g. via bootstrapping) to illustrate when a stable plateau is reached. This is the crucial number of cells that have to be annotated I guess (supposed that the three classes are balanced).*
- *Is 'Trash' the same class as 'Discard' in Fig. 2?*
- *I suggest to use the same color code as in Fig. 2*

A2.1

- Average annotation time was 3.5 sec/annotated sample. Now this is included in Supplementary figure 1.
- We agree with Reviewer 2 here this is indeed a crucial point. We created two new panels for Supplementary figure 1 as proposed by the Reviewer. In the case of nNOS cells, we observed that the accuracy reached a plateau relatively early, while for the other case this increase was somewhat slower. The accuracy drop was due to instability and not real accuracy drop, we are thankful to the reviewer for highlighting this issue.
- 'Discard' and 'Trash' were inconsistent names used to refer to cells we are not interested in. We have now clarified this in the revised manuscript and use 'discard' throughout. In the first experiment, discard refers to cells that are touching or have segmentation errors. In the second experiment it refers to cells with no GFP signal and image analysis errors.
- Now we use the same color codes in case of Figure 2 and Supplementary figure 1. We are grateful to the Reviewer for highlighting these anomalies.

10-fold cross-validation and confusion matrices for a random forest classifier classifying nNOS+/nNOS-/trash (left, n=157) and foci-forming/homogeneous/no-gfp/trash for the last experiment (right, n=165).

R2.2
Presentation

Fig 2 has a number of issues:

- Is 'Discard' in 2b a separate class that is used for training and prediction? For me, these 'Discard' nuclei look rather nice, at least I can see no 'artifacts' as stated in the figure caption. Pls detail why and how they are kicked out.

A2.2

Cells marked as 'Discard' in the top of 2b are most likely apoptotic, while the discarded contours in the bottom right are in close proximity to other cells or are recognized as being part of cell clumps.

R2.3

Presentation

- Fig 2c does not show 'the same region' as 2b. It would be nice though.

A2.3

Figure 2b and Figure 2c show the same region (up to +/- 1Um). We highlight below a few characteristic points that show corresponding cells. The second image was taken several hours after the first one, which may have caused the nNOS+ signal to appear fainter.

R2.4

Explanation

From my perspective, the third experiment is still not properly described:

- 10 different cell lines expressing 10 different shRNAs are mixed and tested for homogeneous vs. foci forming polymerase distribution. Is DNA or RNA of these cell lines sequenced at the end ('Read count ratio')?

A2.4

Single cells displaying either homogeneous or foci-forming polymerase distribution were captured and their DNA was sequenced. We made the following changes in the main text (highlighted in yellow) to clarify the issue:

- "We prepared a mixture of single shRNA-expressing stable human embryonic kidney cell lines (limited to 10 cell lines in our study)."
- "While RNAi knockdowns tests one gene at a time, measuring population responses (~20,000 experiments for a genome-wide library), CAMI technology can be used to select individual cells from a mixture of stably silenced cell lines."

- “The DNA of extracted cells is sequenced using universal primers flanking the specific silencing shRNA-coding region present in each cell of the library.”
- “Using CAMI, we automatically identified 150 foci-forming and 150 homogeneous cells, captured them, and sequenced their shRNA-coding DNA region using NGS.”

R2.5

Explanation

- How do the authors discriminate between the 4 classes shown in suppl fig 1?

A2.5

The machine learning uses extracted image features to predict the phenotype of each cell as either foci-forming, homogeneous, discard (no-GFP), or discard (segmentation error). The overall accuracy of the classifier is approximately 88%, but we only select cells with the most confident phenotypes. In effect, the chance of selecting a cell of the wrong type for sequencing is very small.

R2.6

Explanation

- Does 'Trash' and 'No GFP' in suppl fig 1 correspond to 'Discard' in fig 2g?

A2.6

'Discard' in Figure 2g refers to cells that either have image analysis errors or do not contain GFP signal. These two types of discarded cells are now more clearly represented in Supplementary figure 1.

R2.7

Explanation

- I suggest to order the bars in Fig 2i.

A2.7

This is a great idea we modified Figure 2i accordingly.

R2.8

Statistical analysis

Fig. 2e is supposed to show that selected cells differ in the expression of nNOS. I have a number of issues with this:

- A statistical test should be applied to prove that nNOS expression differs significantly while S18 does not.

A2.8

We have added the following statistical test to the caption of Figure 2e: “Expression levels measured by dPCR show CAMI reliably separates cells. Cells identified as nNOS+ show significantly higher expression (7.96 ± 0.48) than those identified as nNOS- (0.48 ± 0.95), two-sample t-test $p=0.0061$. Expression levels of housekeeping gene S18 did not vary significantly between cells identified as nNOS+ (116.37 ± 16.54) and nNOS- (103.98 ± 10.29), two-sample t-test $p=0.1992$.”

R2.9

Statistical analysis

- Fig 2d shows 6 nNOS negative cells. Only 4 nNOS neg cells are shown in Fig. 2e. Why only that few? Again, to promote the high-throughput potential of the method, all isolated cells should be shown.

A2.9

In the experiment in panel 2e, we only used four nNOS- cells. However, we collected many more nNOS- cells for the experiment in panel 2f. For both experiments, we collected over 400 nNOS- cells. The cells in panel 2d depict a few of the cells selected among the two experiments. Space limitations did not allow for the inclusion of images of all the collected cells.

R2.10

Statistical analysis

- Logarithmic y axis labels are not necessary, neither for the few transcripts of nNOS, neither for the homogenous many transcripts of S18.

A2.10

There is an order of magnitude difference between the number of RNA in nNOS+ interneurons and S18, and nearly a further order of magnitude between nNOS+ interneurons and nNOS- interneurons. We believe that this relationship is most easily visualized using a logarithmic scale.

R2.11

Statistical analysis

- The depicted micro tubes do not convey any information in Fig 2e and 2i, and they are misleading in Fig. 2f

A2.11

The purpose of the micro tube diagrams is to visually convey to the reader whether the cells were collected individually or pooled in each of the experiments. They also help to clarify whether populations were mixed or separated. We believe this is a useful and interesting information for the reader. However, we agree that in Fig 2f the arrangement of the microtubes was misleading, thus, we now updated the panel.

Reviewer #3

R3.1

I agree with the authors that single cell PCR is a fine method for assaying gene expression on a gene by gene basis. My rationale for suggesting including further single cell sequencing data is that it provides a more global view of the transcriptome, which allows for assessment of measures such as gene dropout and library complexity that can be reflective of the quality of single cell libraries and can be valuable for assessing the utility of a given method for downstream applications. Perhaps this is beyond the scope of the study, but to convincingly demonstrate that this laser capture based method is a viable alternative to FACS isolation of cells of interest, I would want to see a more in-depth study of a larger population of single cells.

R3.1 Editor's advice:

While we agree with reviewer 3 that their first point could be considered out of scope, we ask that you address this reviewer's two other points.

A3.1

We believe that Reviewer #3 raises here a very interesting point, however we also think it is beyond the scope of this work.

R3.2

The authors use their single cell PCR data to demonstrate that RNA quality is relatively preserved after fixation, immunohistochemistry, and cell capture. Again, I find that this gives a limited view of the overall RNA quality within the captured cells because transcripts can vary in their rate of degradation and only a few genes were assayed in the present study. It would be helpful to see additional data supporting preservation of high quality RNA within the samples. An estimate of RIN from a small population of captured cells would be helpful. It does appear that there is some evidence of degraded RNA in the cDNA profiles provided as illustrated by the presence of low molecular weight products in the traces.

A3.2

We agree with the Reviewer that single-cell digital PCR data alone cannot fully describe RNA purity and that it does not provide explicit integrity measure after cell capture. In our manuscript we outlined that similar number of transcripts could be detected for a number of genes regardless of the sample used, i.e. no difference could be seen from live-cell aspirations (after electrophysiological measurements) and from fixed, laser captured cells (Supplementary Table 1).

To minimize RNA degradation in the minute amounts of samples we optimized the entire protocol, including sample fixation (optimized time and high quality freshly prepared fixative reagents), single cell collection and processing (among other by using SingleCellProtect reagent). During the optimization we have indeed determined RIN derived from small population of captured cells by using High Sensitivity RNA Screen Tape and Reagents on TapeStation 2100 (all from Agilent). We have always obtained RINe values of >6, which indeed indicate partial RNA degradation; here we show a typical example of such a measurement.

With respect to cDNA, low molecular weight products can be derived from degradation, but also from dimerized primers and linkers, which always appear when low concentration input RNA is used.

During the optimization of both the digital PCR as well as the scRNA-Seq experiments we did realized that partial RNA degradation is indeed present. However, taking into consideration the experimental setup and the number of potentially RNA damaging steps that must be executed, we argue that the presented data are still valuable. We do acknowledge, however, that further fine-tuning of the experimental setup is possible, but the main focus of this work is an image-based single-cell isolation workflow.

R3.3

The authors state that they used a total RNA kit to prepare their pooled cell samples for sequencing because they wanted to extract as much information as possible, but I think that this has limited the amount of information that can be mined from their sequencing data. Without depletion of ribosomal RNA or use of an oligodT based method, the bulk of RNA captured and sequenced will be ribosomal. From the mapping statistics provided, it seems that the % of mapped reads is very low (~25% for neuronal samples and ~17% for astrocyte sample) which likely reflects most of the reads being occupied by sequencing of rRNA, but could also be influenced by sample quality, sequencing quality, and library complexity. Can the authors provide a value for the % of reads mapping to rRNA? The authors then provide a table of the top 100 expressed genes across their different replicates. Can they also provide a summary of the total number of genes detected? It looks like maybe only a few hundred genes were detected given that the list of top expressers includes many genes with low (<5) RPKM values. The authors state that even though they used total RNA for their sequencing experiments, they still mostly detected protein coding genes but the list of top expressing genes includes many uncharacterized and non-coding transcripts. Where does the “99,60 – 99,97% of the detected biotypes fall within protein coding class” statement derive from? For example, for the 50x astrocyte sample, many of the genes expressed are U1 and U6 small nuclear spliceosomal RNAs. Also, it is interesting that many of the common marker genes for pyramidal neurons (SLC17A7) and astrocytes (GFAP, AQP4) are not among the highest expressed genes in the pooled samples. Are these detected at all? One would be hard pressed to guess what type of cells were sequenced given the gene list provided. Is there any expression of genes that do not correspond to the cell class captured in the pooled samples – i.e. expression of MBP or MOG in the neuronal or astrocyte samples? Is there any evidence of contamination from other cell types incidentally captured by the sampling technique?

A3.3

As we have mentioned during both the first revision as well as in our response above, the purpose of the presented proof of principle experiments was to show that biological material obtained with CAMI is suitable – among others – for RNA sequencing. That was the reason why the sequencing was first set to ~2 million reads and later – after receiving the first comments by the reviewers - to ~4 million reads per sample. There is no question that this is a very low sequencing depth from which the only conclusion that can be drawn is that the isolated cells – no matter if 50 or 300 cells are isolated – are suitable for downstream applications such as scRNA-Seq. We think any further discussion would be over-interpretation of the data, which we wanted to avoid.

To obtain biologically relevant sequencing data, the desired sequencing depth would be ~40 million reads per sample. Without further increasing the depth of the sequencings, the answers to the questions raised by the reviewer are as follows:

- as correctly highlighted by the reviewer, the reads mapping to annotated rat genome is indeed low, which is partly because of the rRNA leftover in the samples (furthermore, the annotation of the rat genome is very poor). Another reason is the relatively low-quality input RNA, as a

consequence the resulting cDNA has a broad size range. Taking all these together, it was difficult to setup a proper fragmentation protocol: shorter time applied for fragmentation resulted in sequencing libraries with longer inserts that could not be sequenced due to the sequencing chemistry applied, in contrast, fragmentation for longer times resulted in very short inserts that were filtered out with the trimming protocol (raw reads of shorter than 40 bases were filtered out).

- again, the very low sequencing depth resulted in low RPKM values
- unique exon reads mapping to SLC17A7 and GFAP in pyramidal cells and astrocytes, respectively, are present; however, we could not detect reads mapping to AQP4
- no reads were mapped to MBP nor to MOG in any of the 4 samples which suggest no contamination from oligodendrocytes or myelin
- in the current experimental setup it is very hard, if not impossible, to determine whether contamination from other cell type/s is present, partly because 50 or 300 cells were used and the RNA from a single contaminating cell would probably be titrated out by the rest of the cells.

We emphasize that our experimental setup was designed and conducted as a proof of principle to demonstrate the capabilities of the CAMI technology to target, isolate and analyze individual cells and that we show a strong correlation between cells identified as nNOS+ from different animals. We also demonstrate that each of the replicates show very little correlation with cells identified as astrocytes. We are moving toward an implementation of the full workflow and are aware that additional fine-tunings are necessary in order to be able to draw biologically relevant conclusions. In the process, several questions arise such as 1) whether or not to isolate higher quality RNA, 2) if the REPLI-g WTA Single Cell Kit is the best choice, and 3) if we can go below 50 pooled cells, etc. We are currently working on all these aspects which, once settled, will help answer biologically relevant questions.

REVIEWERS' COMMENTS:

Reviewer #2 (Remarks to the Author):

The authors clarified and clearly improved a number of issues. I still have two points that should be addressed.

1. Supplementary figure 1 shows interesting metrics of the machine learning approach. The presentation of the figure can be improved though:

- Please give all values in the confusion matrix (ideally with proper rounding).
- Please specify what is 'true' and what is 'predicted' in the matrix.
- Please provide proper figure captions, e.g. detailing how the error bands in the lower plots have been calculated.

2. I still have issues with Fig. 2e

- The authors state in their response: 'In the experiment in panel 2e, we only used four nNOS- cells. However, we collected many more nNOS- cells for the experiment in panel 2f. For both experiments, we collected over 400 nNOS- cells. The cells in panel 2d depict a few of the cells selected among the two experiments. Space limitations did not allow for the inclusion of images of all the collected cells.' If the authors collected and measured hundreds of single cells, they should show all the expression values in Fig 2e (not all the images in 2d of course). As a suggestion, beeswarm plots are a nice way to show all data points and do not require much space.
- The authors now show that nNOS is significantly differentially expressed in the two populations. However, it is not clear from the figure caption if this test has been performed on all hundreds of cell (which should be done), or on only the few examples shown. Please provide the number of cells used for the test in the caption to clarify this for the reader.
- I suggest to make the axis label more explicit, e.g. 'Number of nNOS RNA'.

Reviewer #3 (Remarks to the Author):

I believe that the authors have sufficiently addressed all of my previous comments and recommend acceptance of this version of the paper.

Detailed response to Reviewer 2

In the following, the comments from the reviewer have been enumerated. Our response immediately follows each comment. The first comment from Reviewer #2 is denoted R2.1, and our response is denoted A2.1.

Reviewer #2 (Remarks to the Author):

The authors clarified and clearly improved a number of issues. I still have two points that should be addressed.

[R2.1]

1. Supplementary figure 1 shows interesting metrics of the machine learning approach. The presentation of the figure can be improved though:

- Please give all values in the confusion matrix (ideally with proper rounding).

[A2.1]

We agree with this suggestion and made the proposed changes accordingly.

[R2.2]

- Please specify what is 'true' and what is 'predicted' in the matrix.

[A2.2]

This suggestion has been added to the figure now.

[R2.3]

- Please provide proper figure captions, e.g. detailing how the error bands in the lower plots have been calculated.

[A2.3]

The above text is now in the panel description:

“A randomly selected set of annotated samples were used for cross validation. Error bars depict the mean and standard deviation of cross validation accuracies (n=100).”

[R2.4]

2. I still have issues with Fig. 2e

- The authors state in their response: 'In the experiment in panel 2e, we only used four nNOS- cells. However, we collected many more nNOS- cells for the experiment in panel 2f. For both experiments, we collected over 400 nNOS- cells. The cells in panel 2d depict a few of the cells selected among the two experiments. Space limitations did not allow for the inclusion of images of all the collected cells.' If the authors collected and measured hundreds of single cells, they should show all the expression values in Fig 2e (not all the images in 2d of course). As a suggestion, beeswarm plots are a nice way to show all data points and do not require much space.

[A2.4]

Probably this reviewer misunderstood our comment, we did not perform single cell digital PCR on hundreds of cells, but only on those that are presented. We performed though other molecular analysis (transcriptomics) on the larger amount of cells of the same phenotype.

[R2.5]

- The authors now show that nNOS is significantly differentially expressed in the two populations. However, it is not clear from the figure caption if this test has been performed on all hundreds of cell (which should be done), or on only the few examples shown. Please provide the number of cells used for the test in the caption to clarify this for the reader.

[A2.5]

The significance test was performed only on those cells that were used for the dPCR (11 cells).

[R2.6]

- I suggest to make the axis label more explicit, e.g. 'Number of nNOS RNA'.

[A2.6]

We agree with this suggestion and changed the label accordingly.

Reviewer #3 (Remarks to the Author):

I believe that the authors have sufficiently addressed all of my previous comments and recommend acceptance of this version of the paper.